# DATA-EFFICIENT AUGMENTATION FOR TRAINING NEURAL NETWORKS

## ABSTRACT

Data augmentation is essential to achieve state-of-the-art performance in many deep learning applications. However, modern data augmentation techniques become computationally prohibitive for large datasets. To address this, we propose a rigorous technique to select subsets of data points that when augmented, closely capture the training dynamics of full data augmentation. We first show that data augmentation, modeled as additive perturbations, speeds up learning by enlarging the smaller singular values of the network Jacobian. Then, we propose a framework to iteratively extract small subsets of training data that when augmented, closely capture the alignment of the fully augmented Jacobian with label/residual vector. We prove that stochastic gradient descent applied to augmented subsets found by our approach have similar training dynamics to that of fully augmented data. Our experiments demonstrate that our method outperforms state-of-the-art max-loss strategy by 7.7% on CIFAR10 while achieving 6.3x speedup, and by 4.7% on SVHN while achieving 2.2x speedup, using 10% and 30% subsets, respectively.

## 1 INTRODUCTION

Data augmentation expands the training data by applying transformations, such as rotations or crops for images, to the original training examples. Due to its effectiveness, data augmentation is a key component in achieving nearly all state-of-the-art results in deep learning applications (Shorten & Khoshgoftaar, 2019). The most effective data augmentation techniques often search over a (possibly large) space of transformations to find sequences of transformations that speeds up training the most (Cubuk et al., 2019; 2020; Luo et al., 2020; Wu et al., 2020). In addition, multiple augmented examples are usually generated for a single data point to obtain better results, increasing the size of the training data by orders of magnitude. As a result, state-of-the-art data augmentation techniques become computationally prohibitive for large real-world problems (*c.f.* Fig. 1).

To make data augmentation more efficient and scalable, an effective approach is to carefully select a small subset of the training data such that augmenting only the subset have similar training dynamics to that of full data augmentation. If such a subset can be quickly found, it would directly lead to a significant reduction in storage and training costs, and lower costs incurred from selecting and tuning the optimal set of transformations to apply. Despite the efficiency and scalability that it can provide, this direction has remained largely unexplored. Existing studies are limited to fully training a network and subsampling data points based on its loss or influence for augmentation in subsequent training runs (Kuchnik & Smith, 2018). However, this method is prohibitive for large datasets, provides

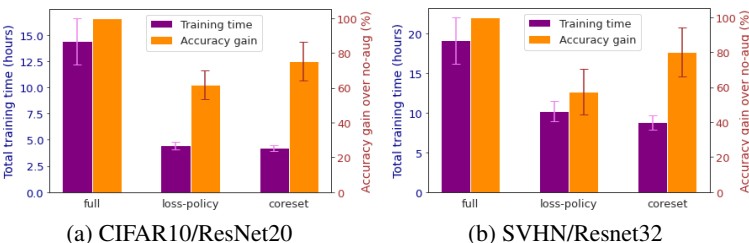

(a) CIFAR10/ResNet20          (b) SVHN/Resnet32

Figure 1: Speedup/Accuracy of augmenting 30% coresets compared to original max-loss policy for (a) ResNet20 trained on CIFAR10 and (b) ResNet32 trained on SVHN.

a marginal improvement over augmenting random subsets, and does not provide any theoretical guarantee for the performance of the network trained on the augmented subsets.

A major challenge in finding the most effective data points for augmentation is to theoretically understand how data augmentation affects the optimization and generalization of neural networks. Existing theoretical results are mainly limited to simple linear classifiers and analyze data augmentation as enlarging the span of the training data (Wu et al., 2020), providing a regularization effect (Bishop, 1995; Dao et al., 2019; Wager et al., 2013; Wu et al., 2020), enlarging the margin of a linear classifier (Rajput et al., 2019), or having a variance reduction effect (Chen et al., 2019). However, such tools do not provide insights on the effect of data augmentation on training deep neural networks.

Here, we study the effect of label invariant data augmentation modeled by small additive perturbations (Rajput et al., 2019) on training dynamics of overparameterized neural networks. In particular, we rely on recent results that characterize training dynamics of neural networks based on the alignment of the labels/residuals with the singular subspace of the network's Jacobian matrix containing all its first-order partial derivatives (Arora et al., 2019). We show that label invariant data augmentation enlarges smaller singular values of the information space, and prove that this will speed up training.

Next, we develop a rigorous method to iteratively find small weighted subsets (coresets) that when augmented, closely capture the alignment of the full augmented data with the label/residual, at every point during the training. Augmenting the coresets guarantees similar training dynamics to that of full data augmentation. Our key observation is that early in training, this alignment is best captured by data points that are highly representative of their classes. However towards the end of training when the network converges, data points with maximum loss best capture this alignment. Data augmentation has been empirically shown to mainly affect the initial phase of training (Golatkar et al., 2019), which crucially determines the final basin of convergence (Fort et al., 2020). Better selection of points by our method during this initial phase explains the superior accuracy improvement resulted by augmenting our coresets vs. the max-loss and the better final generalization performance. Importantly, we show that our coresets can be provably trained on even in absence of full data, and even when a high fraction of labels are noisy.

We demonstrate the effectiveness of our approach applied to training ResNet20, ResNet32, Wide-ResNet on CIFAR10, CIFAR10-IB and SVHN, compared to random and max-loss baselines (Kuchnik & Smith, 2018). We show that augmenting coresets found by our approach outperforms the state-of-the-art even in absence of the full data. For instance, when small augmented subsets of size $30\%$ found by our approach are appended to CIFAR10, we attain $75\%$ of the improvement in test accuracy compared to augmenting the full dataset while enjoying a 3.4x speedup in training time. Even in the absence of full data, training and augmenting tiny coresets of size $1\%$ can achieve $74.7\%$ accuracy on CIFAR10/ResNet20 while providing a 39x speedup compared to using the full dataset. On CIFAR10 with $50\%$ noisy labels, augmenting $50\%$ of the training data outperforms full data augmentation.

## 2 PROBLEM FORMULATION

We begin by formally describing the problem of learning from augmented data. Consider a dataset $\mathcal{D}_{train} = (\boldsymbol{X}_{train}, \boldsymbol{y}_{train})$, where $\boldsymbol{X}_{train} = (\boldsymbol{x}_1, \cdots, \boldsymbol{x}_n) \in \mathbb{R}^{d \times n}$ is the set of $n$ normalized data points $\boldsymbol{x}_i \in [0,1]^d$, from the index set $V$, i.e., $i \in V = \{1, \cdots, n\}$, and $\boldsymbol{y}_{train} = (y_1, \cdots, y_n) \in \{y \in \{\nu_1, \nu_2, \cdots, \nu_C\}\}$ with $\{\nu_j\}_{j=1}^C \in [0,1]$. Assume at every step $t$ during training, we have a set of augmented examples $\mathcal{D}_{aug}^t$ generated by a set of label-invariant transformations. In particular, following (Rajput et al., 2019) we model data augmentation as an arbitrary bounded additive perturbation $\boldsymbol{\epsilon}$, with $\|\boldsymbol{\epsilon}\| \leq \epsilon_0$. Formally, for a given upper bound $\epsilon_0$ and the set of all possible transformations $\mathcal{A}$, we study the transformations selected from $\mathcal{S} \subseteq \mathcal{A}$ satisfying

$$\mathcal{S} = \{T_i \in \mathcal{A} \mid \|T_i(\boldsymbol{x}) - \boldsymbol{x}\| \leq \epsilon_0 \ \forall \boldsymbol{x} \in \boldsymbol{X}^{train}\}. \tag{1}$$

Under the smoothness constraint of images, where adjacent pixels have close intensities, such as small translations, crops, rotations, and for other pixel-wise augmentation methods such as sharpening, blurring, and color distortions (Cubuk et al., 2020), $\epsilon_0$ is small. This model is also especially suitable for modelling state-of-the-art augmentation techniques such as structured adversarial perturbation (Luo et al., 2020), in which pixel intensities are changed minimally. While in our analysis we focus on small perturbations $\varepsilon_0$, our experiments use variety of strong and weak augmentations. Although other transformations such as data synthesis (Baluja & Fischer, 2017; Mirza & Osindero, 2014),

semantic augmentation (Wang et al., 2019), large translations, crops, rotations, and flips are still valid under the additive perturbation model, they can be more effectively modelled using matrices of linear transforms (Wu et al., 2020), as we analyze in Appendix B.3.

In practice, multiple augmentations are generated for each example $\boldsymbol{x}_i$, and each augmented data point can be a combination of multiple transformations, e.g. random cropping and rotating followed by horizontal flipping. The set of augmentations at iteration $t$ generating $r$ augmented examples per data point can be specified, with abuse of notation, as $\mathcal{D}_{aug}^t = \{\bigcup_{i=1}^r (T_i^t(\boldsymbol{X}_{train}), \boldsymbol{y}_{train})\}$, where $|\mathcal{D}_{aug}^t| = rn$ and $T_i^t(\boldsymbol{X}_{train})$ transforms all the training data points with the set of transformations $T_i^t \subset \mathcal{S}$ at iteration $t$. We denote $\boldsymbol{X}_{aug}^t = \{\bigcup_{i=1}^r T_i^t(\boldsymbol{X}_{train})\}$ and $\boldsymbol{y}_{aug}^t = \{\bigcup_{i=1}^r \boldsymbol{y}_{train}\}$.

Let $f(\boldsymbol{W}, \boldsymbol{x})$ be an arbitrary neural network with $m$ vectorized (trainable) parameters $\boldsymbol{W} \in \mathbb{R}^m$. We assume that the network is trained using (stochastic) gradient descent with learning rate $\eta$ to minimize the squared loss $\mathcal{L}$ over the original and augmented training examples $\mathcal{D}^t = \{\mathcal{D}_{train} \cup \mathcal{D}_{aug}^t\}$ with associated index set $V^t$, at every iteration $t$:

$$\mathcal{L}(\boldsymbol{W}^t, \boldsymbol{X}) := \frac{1}{2} \sum_{i \in V^t} \mathcal{L}_i(\boldsymbol{W}^t, \boldsymbol{x}_i) := \frac{1}{2} \sum_{(\boldsymbol{x}_i, y_i) \in \mathcal{D}^t} \|f(\boldsymbol{W}^t, \boldsymbol{x}_i) - y_i\|_2^2. \tag{2}$$

The gradient update at iteration $t$ is given by

$$\boldsymbol{W}^{t+1} = \boldsymbol{W}^t - \eta \nabla \mathcal{L}(\boldsymbol{W}^t, \boldsymbol{X}), \quad \nabla \mathcal{L}(\boldsymbol{W}^t, \boldsymbol{X}) = \mathcal{J}^T(\boldsymbol{W}^t, \boldsymbol{X})(f(\boldsymbol{W}^t, \boldsymbol{X}) - \boldsymbol{y}), \tag{3}$$

where $\boldsymbol{X}^t = \{\boldsymbol{X}_{train} \cup \boldsymbol{X}_{aug}^t\}$ and $\boldsymbol{y}^t = \{\boldsymbol{y}_{train} \cup \boldsymbol{y}_{aug}^t\}$ are the set of original and augmented examples and their labels, $\mathcal{J}(\boldsymbol{W}, \boldsymbol{X}) \in \mathbb{R}^{n \times m}$ is the Jacobian matrix associated with $f$, and $\boldsymbol{r}^t = f(\boldsymbol{W}^t, \boldsymbol{X}) - \boldsymbol{y}$ is the residual. We further assume that $\mathcal{J}$ is smooth with Lipschitz constant $L$:

$$\|\mathcal{J}(\boldsymbol{W}, \boldsymbol{x}_i) - \mathcal{J}(\boldsymbol{W}, \boldsymbol{x}_j)\| \le L \|\boldsymbol{x}_i - \boldsymbol{x}_j\| \qquad \forall \, \boldsymbol{x}_i, \boldsymbol{x}_j \in \boldsymbol{X}. \tag{4}$$

This trivially holds for linear models, and when $\boldsymbol{W}$ is bounded, it holds for deep ReLU, and generally for networks with any activation $\phi$ with bounded derivatives $\phi'$ and $\phi''$ (Jordan & Dimakis, 2020). Under this assumption, augmentation as defined in Eq. (1) results in bounded perturbations to the Jacobian matrix. I.e., for any transformation $T_j \in \mathcal{S}$, we have $\|\mathcal{J}(\boldsymbol{W}, \boldsymbol{x}_i) - \mathcal{J}(\boldsymbol{W}, T_j(\boldsymbol{x}_i))\| \le L\epsilon_0$. Using the shorthand notations $\mathcal{J} = \mathcal{J}(\boldsymbol{W}, \boldsymbol{X}_{train})$ and $\tilde{\mathcal{J}} = \mathcal{J}(\boldsymbol{W}, T_j(\boldsymbol{X}_{train}))$, we obtain $\tilde{\mathcal{J}} = \mathcal{J} + \boldsymbol{E}$, where $\boldsymbol{E}$ is the perturbation matrix with $\|\boldsymbol{E}\|_2 \le \|\boldsymbol{E}\|_F \le \sqrt{n}L\epsilon_0$.

## 3  DATA AUGMENTATION SPEEDS UP LEARNING

In this section, we analyze the effect of data augmentation on training dynamics of neural networks, and show that data augmentation can provably speed up learning. To do so, we leverage the recent results that characterize the training dynamics based on properties of neural network Jacobian matrix and the corresponding Neural Tangent Kernel (NTK) (Jacot et al., 2018) defined as $\boldsymbol{\Theta} = \mathcal{J}(\boldsymbol{W}, \boldsymbol{X})\mathcal{J}(\boldsymbol{W}, \boldsymbol{X})^T$. Formally (Arora et al., 2019):

$$\boldsymbol{r}^t = \sum_{i=1}^n (1 - \eta\lambda_i)(\boldsymbol{u}_i \boldsymbol{u}_i^T)\boldsymbol{r}^{t-1} = \sum_{i=1}^n (1 - \eta\lambda_i)^t (\boldsymbol{u}_i \boldsymbol{u}_i^T)\boldsymbol{r}^0, \tag{5}$$

where $\boldsymbol{\Theta} = \boldsymbol{U}\boldsymbol{\Lambda}\boldsymbol{U}^T = \sum_{i=1} \lambda_i \boldsymbol{u}_i \boldsymbol{u}_i^T$ is the eigendecomposition of the NTK. Although the constant NTK assumption holds only in the infinite width limit, Lee et al. (2019) found close empirical agreement between the NTK dynamics and the true dynamics for wide but practical networks, such as wide ResNet architectures (Zagoruyko & Komodakis, 2016). Eq. (5) shows that training dynamics depend on the alignment of the NTK with the residual vector at every iteration $t$. In particular, shrinkage of residuals along the directions associated with larger eigenvalues of the NTK is fast and happens early during the training, while learning along the space associated with the small eigenvalues is slow and happens later. In the following, we prove that for small perturbations $\epsilon_0$, data augmentation speeds up training by enlarging smaller eigenvalues of the NTK, while decreasing larger eigenvalues with a high probability. Intuitively, this can be characterized as decreasing the learning rate for dimensions with larger gradient and slightly increasing the learning in dimensions with smaller gradients, and having a regularization effect by slightly perturbing the eigenvectors.

(a) MNIST/MLP Epoch 15 (b) C10/Res20 Epoch 15 (c) SVHN/Res32 Epoch 15 (d) MNIST/MLP Epoch 0

Figure 2: Histogram of the singular values of the Jacobian matrix for augmented vs. original (a) MNIST/1 hidden-layer MLP, (b) CIFAR10/ResNet20, and (c) SVHN/ResNet32. (d) Effect of strong vs. weak augmentation on MNIST/MLP at initialization. We sampled 2 and 4 transformations from rotation, translation, contrast, brightness, etc. for normal vs. strong augmentation.

### 3.1 EFFECT OF DATA AUGMENTATION ON THE EIGENVALUES OF THE NTK

We first investigate the effect of data augmentation on the singular values of the Jacobian, and use this result to bound the change in the eigenvalues of the NTK. To characterise the effect of data augmentation on singular values of the perturbed Jacobian $\tilde{\mathcal{J}}$, we rely on Weyl's theorem (Weyl, 1912) stating that under bounded perturbations $\boldsymbol{E}$, no singular value can move more than the norm of the perturbations. Formally, $|\tilde{\sigma}_i - \sigma_i| \leq \|\boldsymbol{E}\|_2$, where $\tilde{\sigma}_i$ and $\sigma_i$ are the singular values of the perturbed and original Jacobian respectively. Crucially, data augmentation affects larger and smaller singular values differently. Let $\boldsymbol{P}$ be orthogonal projection onto the column space of $\mathcal{J}^T$, and $\boldsymbol{P}_\perp = \boldsymbol{I} - \boldsymbol{P}$ be the projection onto its orthogonal complement subspace. Then, the singular values of the perturbed Jacobian $\tilde{\mathcal{J}}^T$ are $\tilde{\sigma}_i^2 = (\sigma_i + \mu_i)^2 + \zeta_i^2$, where $|\mu_i| \leq \|\boldsymbol{P}\boldsymbol{E}\|_2$, and $\sigma_{\min}(\boldsymbol{P}_\perp \boldsymbol{E}) \leq \zeta_i \leq \|\boldsymbol{P}_\perp \boldsymbol{E}\|_2$, $\sigma_{\min}$ the smallest singular value of $\mathcal{J}^T$ (Stewart, 1979). Since the eigenvalues of the projection matrix $\boldsymbol{P}$ are either 0 or 1, as the number of dimensions $m$ grows, for bounded perturbations we get that on average $\mu_i^2 = \mathcal{O}(1)$ and $\zeta_i^2 = \mathcal{O}(m)$. Thus, the second term dominates and increase of small singular values under perturbation is proportional to $\sqrt{m}$. However, for larger singular values, first term dominates and hence $\tilde{\sigma}_i - \sigma_i \cong \mu_i$. Thus in general, small singular values can become significantly and proportionally larger, while larger singular values remain relatively unchanged. Empirically, we observed that largest singular values often decrease by data augmentation.

Fig. 2 shows the effect of data augmentation on singular values of the Jacobian matrix for a 1 hidden layer MLP trained on MNIST, ResNet32 trained on SVHN, and ResNet20 trained on CIFAR10. As calculating the entire Jacobian spectrum is computationally prohibitive, data is subsampled from 3 classes. For all datasets, we used the data augmentation techniques described in the Experiments section, such as rotation, translation, contrast, brightness. It can be observed that for less diverse datasets such as MNIST that have smaller singular values in general, data augmentation can easily enlarge the singular values. On the other hand, for more diverse datasets such as CIFAR10 that generally have larger singular values, augmentation cannot change the singular values considerably. In both cases largest singular values became smaller.

The following Lemma characterizes the *expected* change to the eignvalues of the NTK.

**Lemma 3.1** *Data augmentation as additive perturbations bounded by small $\epsilon_0$ results in the following expected change to the eigenvalues of the NTK:*

$$\mathbb{E}[\tilde{\lambda}_i] = \mathbb{E}[\tilde{\sigma}_i^2] = \sigma_i^2 + \sigma_i(1 - 2p_i)\|\boldsymbol{E}\| + \|\boldsymbol{E}\|^2/3 \tag{6}$$

*where $p_i := \mathbb{P}(\tilde{\sigma}_i - \sigma_i < 0)$ is the probability that $\sigma_i$ decreases as a result of data augmentation.*

All the proofs can be found in the Appendix.

From Lemma 3.1 and Eq. (5) we see that augmentation can speed up learning at every epoch by $\sigma_i(1 - 2p_i)\|\boldsymbol{E}\| + \|\boldsymbol{E}\|^2/3$ along dimensions with smaller singular values for which $p_i \leq 0.5$, and by $\|\boldsymbol{E}\|^2/3$ along dimensions with for which singular values do not change ($p_i \approx 0.5$). It can also speed up learning along dimensions for which singular values decrease ($p_i > 0.5$) as long as $\|\boldsymbol{E}\| \geq 3\sigma_i(2p_i - 1)$. This is mainly relevant for smaller singular values under strong augmentation.

### 3.2 EFFECT OF DATA AUGMENTATION ON THE EIGENVECTORS OF THE NTK

Next, we focus on characterizing the effect of data augmentation on the eigenspace of the NTK. Let the singular subspace decomposition of the Jacobian be $\mathcal{J} = \boldsymbol{U}\boldsymbol{\Sigma}\boldsymbol{V}^T$. Then for the NTK, we

have $\boldsymbol{\Theta} = \mathcal{J}\mathcal{J}^T = \boldsymbol{U}\boldsymbol{\Sigma}\boldsymbol{V}^T\boldsymbol{V}\boldsymbol{\Sigma}\boldsymbol{U}^T = \boldsymbol{U}\boldsymbol{\Sigma}^2\boldsymbol{U}^T$ (since $\boldsymbol{V}^T\boldsymbol{V} = \boldsymbol{I}$). Hence, the perturbation of the eigenspace of the NTK is the same as perturbation of the left singular subspace of the Jacobian $\mathcal{J}$. Suppose $\sigma_i$ are singular values of the Jacobian. Let the perturbed Jacobian be $\tilde{\mathcal{J}} = \mathcal{J} + \boldsymbol{E}$, and denote the eigengap $\gamma_0 = \min\{\sigma_i - \sigma_{i+1} : i = 1, \cdots, r\}$ where $\sigma_{r+1} := 0$. Assuming $\gamma_0 \geq 2\|\boldsymbol{E}\|_2$, a combination of Wedin's theorem (Wedin, 1972) and Mirsky's inequality (Mirsky, 1960) (the counterpart of Weyl's inequality (Weyl, 1912) for singular values) implies that

$$\|\boldsymbol{u}_i - \tilde{\boldsymbol{u}}_i\| \leq 2\sqrt{2}\|\boldsymbol{E}\|/\gamma_0 \tag{7}$$

This result provides an upper-bound on the change of every left singular vectors of the Jacobian.

However as we discuss below, data augmentation affects larger and smaller singular directions differently. To see the effect of data augmentation on every singular vectors of the Jacobian, let the subspace decomposition of Jacobian be $\mathcal{J} = \boldsymbol{U}\boldsymbol{\Sigma}\boldsymbol{V}^T = \boldsymbol{U}_s\boldsymbol{\Sigma}_s\boldsymbol{V}_s^T + \boldsymbol{U}_n\boldsymbol{\Sigma}_n\boldsymbol{V}_n^T$, where $\boldsymbol{U}_s$ associated with nonzero singular values, spans the column space of $\mathcal{J}$, which is also called the signal subspace, and $\boldsymbol{U}_n$, associated with zero singular values ($\boldsymbol{\Sigma}_n = 0$), spans the orthogonal space of $\boldsymbol{U}_s$, which is also called the noise subspace. Similarly, let the subspace decomposition of the perturbed Jacobian be $\tilde{\mathcal{J}} = \tilde{\boldsymbol{U}}\tilde{\boldsymbol{\Sigma}}\tilde{\boldsymbol{V}}^T = \tilde{\boldsymbol{U}}_s\tilde{\boldsymbol{\Sigma}}_s\tilde{\boldsymbol{V}}_s^T + \tilde{\boldsymbol{U}}_n\tilde{\boldsymbol{\Sigma}}_n\tilde{\boldsymbol{V}}_n^T$, and $\tilde{\boldsymbol{U}}_s = \boldsymbol{U}_s + \Delta\boldsymbol{U}_s$, where $\Delta\boldsymbol{U}_s$ is the perturbation of the singular vectors that span the signal subspace. Then the following general first-order expression for the perturbation of the orthogonal subspace due to perturbations of the Jacobian characterize the change of the singular directions: $\Delta\boldsymbol{U}_s = \boldsymbol{U}_n\boldsymbol{U}_n^T\boldsymbol{E}\boldsymbol{V}_s\boldsymbol{\Sigma}_s^{-1}$ (Li et al., 1993). We see that while singular vectors associated to larger singular values are more robust to data augmentation, singular vectors associated with small singular values are more affected. We also note that singular vectors are more robust to perturbations than singular values.

### 3.3 Augmentation Improves Training and Generalization

As discussed, data augmentation increases the smaller eigenvalues of the NTK with a high probability, and is likely to slightly decrease the largest eigenvalues. Besides, eigenvectors (in particular those associated with larger eigenvalues) are generally more robust than eigenvalues. The following Theorem characterizes the expected improvement in the training dynamics resulted by data augmentation.

**Theorem 3.2** *Gradient descent with learning rate $\eta$ applied to a neural network with constant NTK and Lipschitz constant $L$, and data points $\mathcal{D}_{aug}$ augmented with $r$ additive perturbations bounded by $\epsilon_0$ as defined in Eq.* (1) *results in the following training dynamics:*

$$\mathbb{E}[\|\boldsymbol{y} - f(\boldsymbol{X}_{aug}, \boldsymbol{W}^t)\|_2] \leq \sqrt{n\sum_{i=1}^n \left(1 - \eta\left(r\sigma_i^2 + \sqrt{r}\sigma_i(1-2p_i)\|\boldsymbol{E}\| + \|\boldsymbol{E}\|^2/3\right)\right)^{2t}} \tag{8}$$

*where $\boldsymbol{E}$ with $\|\boldsymbol{E}\| \leq \sqrt{n}L\epsilon_0$ is the perturbation to the Jacobian, and $p_i := \mathbb{P}(\tilde{\sigma}_i - \sigma_i < 0)$ is the probability that $\sigma_i$ decreases as a result of data augmentation.*

**Corollary 3.3** *Under the same assumptions as in Theorem 3.2 let $\sigma_{\min}$ be the minimum singular value of Jacobian $\mathcal{J}$ associated with training data $\mathcal{X}_{train}$, then probability $1 - \delta$ generalization error of the network trained with gradient descent on augmented data $\boldsymbol{X}_{aug}$ enjoys the following bound:*

$$\sqrt{\frac{2}{(\sigma_{\min} + \sqrt{n}L\epsilon_0)^2}} + \mathcal{O}\left(\log\frac{1}{\delta}\right). \tag{9}$$

## 4 Most Effective Subsets for Data Augmentation

In Sec. 3 we discussed the effect of data augmentation on changing the alignment of the NTK with the residual, and how it improves training. Here, we focus on identifying the most effective subsets for data augmentation. Our key idea is to find subsets of data points that when augmented, closely capture the alignment of the NTK (or equivalently the Jacobian) corresponding to the full augmented data with the residual vector, $\mathcal{J}(\boldsymbol{W}^t, \boldsymbol{X}_{aug}^t)^T\boldsymbol{r}_{aug}^t$. If such subsets can be found, augmenting only the subsets will change the NTK and its alignment with the residual in a similar way as that of full data augmentation, and will result in similar improved training dynamics. Effectively, such augmented subsets can be trained on along with the full non-augmented data, or without it to further

---

**Algorithm 1** CORESETS FOR EFFICIENT DATA AUGMENTATION

---

**Input:** The dataset $\mathcal{D} = \{(\boldsymbol{x}_i, y_i)\}_{i=1}^n$, number of iterations $T$.
**Output:** Output model parameters $\boldsymbol{W}^T$.
1: **for** $t = 1, \cdots, T$ **do**
2:      $S^t = \emptyset, \boldsymbol{X}_{aug}^t = \emptyset$.
3:      **for** $c \in \{1, \cdots, C\}$ **do**
4:          $S_c^t = \text{greedy}(V_c)$                $\triangleright$ Extract a coreset from class $c$ by solving Eq. (11)
5:          $\gamma_j = \sum_{i \in V_c} \mathbb{1}[j = \arg\min_{j' \in S} \|\mathcal{J}^T(\boldsymbol{W}^t, \boldsymbol{x}_i)r_i - \mathcal{J}^T(\boldsymbol{W}^t, \boldsymbol{x}_{j'})r_{j'}\|]$   $\triangleright$ Coreset weights
6:          $\boldsymbol{X}_{aug}^t = \{\boldsymbol{X}_{aug} \cup \{\cup_{i=1}^r T_i^t(\boldsymbol{X}_{S_c^t})\}\}$             $\triangleright$ Augment the coreset
7:          $\boldsymbol{\rho}_j^t = \gamma_j^t / r$
8:      Update the parameters $\boldsymbol{W}^t$ using weighted gradient descent on $\boldsymbol{X}_{aug}^t$ or $\{\boldsymbol{X}_{train} \cup \boldsymbol{X}_{aug}^t\}$.

---

speed up training. However, generating the full set of transformations $\boldsymbol{X}_{aug}^t$ is often very expensive. This is exacerbated in the case of strong augmentations. Hence, generating the transformations, and then extracting the subsets may not provide a considerable overall speedup.

In the following, we show that weighted subsets (coresets) $S$ that closely estimate the alignment of the Jacobian associated to the original data with the residual vector $\mathcal{J}^T(\boldsymbol{W}^t, \boldsymbol{X}_{train})\boldsymbol{r}_{train}$ can closely estimate the alignment of the Jacobian of the full augmented data and the corresponding residual $\mathcal{J}^T(\boldsymbol{W}^t, \boldsymbol{X}_{aug}^t)\boldsymbol{r}_{aug}^t$. Thus, the most effective subsets for augmentation can be directly found from the training data. Formally, subsets $S_*^t$ weighted by $\boldsymbol{\gamma}_S^t$ that capture the alignment of the full Jacobian with residual by and error of at most $\xi$ can be found by solving the following optimization problem:

$$S_*^t = \underset{S \subseteq V}{\arg\min} |S| \qquad \text{s.t.} \qquad \|\mathcal{J}^T(\boldsymbol{W}^t, \boldsymbol{X}^t)\boldsymbol{r}^t - \text{diag}(\boldsymbol{\gamma}_S^t)\mathcal{J}^T(\boldsymbol{W}^t, \boldsymbol{X}_S^t)\boldsymbol{r}_S^t\| \leq \xi. \tag{10}$$

Solving the above optimization problem is NP-hard. However, as we discuss in the Appendix A.6, a near optimal subset can be found by minimizing the Frobenius norm of a matrix $\boldsymbol{G}_S$, in which the $i^{th}$ row contains the euclidean distance between data point $i$ and its closest element in the subset $S$, in the gradient space. Formally, $[\boldsymbol{G}_S]_{i.} = \min_{j' \in S} \|\mathcal{J}^T(\boldsymbol{W}^t, \boldsymbol{x}_i)r_i - \mathcal{J}^T(\boldsymbol{W}^t, \boldsymbol{x}_{j'})r_{j'}\|$. Intuitively, such subsets contains the set of medoids of the dataset in the gradient space, where the medoids of a dataset are defined as the most centrally located elements in the dataset (Kaufman et al., 1987). The weight of every element $j \in S$ is the number of data points closest to it in the gradient space, i.e., $\gamma_j = \sum_{i \in V} \mathbb{1}[j = \arg\min_{j' \in S} \|\mathcal{J}^T(\boldsymbol{W}^t, \boldsymbol{x}_i)r_i - \mathcal{J}^T(\boldsymbol{W}^t, \boldsymbol{x}_{j'})r_{j'}\|]$. The set of medoids can be found by solving the following *submodular*[1] cover problem:

$$S_*^t = \arg\min_{S \subseteq V} |S| \qquad s.t. \qquad C - \|\boldsymbol{G}_S\|_F \geq C - \xi, \tag{11}$$

where $C \geq \|\boldsymbol{G}_S\|_F$ is a constant. The classical greedy algorithm provides a logarithmic approximation for the above submodular maximization problem, i.e., $|S| \leq (1 + Ln(n))$. It starts with the empty set $S_0 = \emptyset$, and at each iteration $\tau$, it selects the training example $e \in V$ that maximizes the marginal gain $F(e|S_\tau) = F(S_\tau \cup \{e\}) - F(S_\tau)$. Formally, $S_\tau = S_{\tau-1} \cup \{\arg\max_{e \in V} F(e|S_{\tau-1})\}$. The $\mathcal{O}(nk)$ computational complexity of the greedy algorithm can be reduced to $\mathcal{O}(n)$ using randomized methods (Mirzasoleiman et al., 2015), and further improved using lazy evaluation (Minoux, 1978) and distributed implementations (Mirzasoleiman et al., 2013). The rows of the matrix $\boldsymbol{G}$ can be efficiently upper-bounded using the gradient of the loss w.r.t. the input to the last layer of the network, which has been shown to capture the variation of the gradient norms closely (Katharopoulos & Fleuret, 2018). The above upper-bound is only marginally more expensive than calculating the value of the loss, and hence the subset can be found efficiently. Better approximations can also be obtained by considering earlier layers in addition to the last two, at the expense of greater computational cost.

At every iteration $t$ during training, we select a coreset from every class $c \in [C]$ separately, and apply the set of transformations $\{T_i^t\}_{i=1}^r$ only to the elements of the coresets, i.e., $X_{aug}^t = \{\cup_{i=1}^r T_i^t(\boldsymbol{X}_{S^t})\}$. We divide the weight of every element $j$ in the coreset equally among its transformations, i.e. the final weight $\rho_j^t = \gamma_j^t / r$ if $j \in S^t$. We apply the gradient descent updates in Eq. (3) to the weighted Jacobian matrix of $\boldsymbol{X}^t = \boldsymbol{X}_{aug}^t$ or $\boldsymbol{X}^t = \{\boldsymbol{X}_{train} \cup \boldsymbol{X}_{aug}^t\}$ (viewing $\boldsymbol{\rho}^t$ as $\boldsymbol{\rho}^t \in \mathbb{R}^n$) as follows:

$$\boldsymbol{W}^{t+1} = \boldsymbol{W}^t - \eta \left(\text{diag}(\boldsymbol{\rho}^t)\mathcal{J}(\boldsymbol{W}^t, \boldsymbol{X}^t)\right)^T \boldsymbol{r}^t. \tag{12}$$

---

[1]A set function $F : 2^V \to \mathbb{R}^+$ is submodular if $F(S \cup \{e\}) - F(S) \geq F(T \cup \{e\}) - F(T)$, for any $S \subseteq T \subseteq V$ and $e \in V \setminus T$. $F$ is *monotone* if $F(e|S) \geq 0$ for any $e \in V \setminus S$ and $S \subseteq V$.

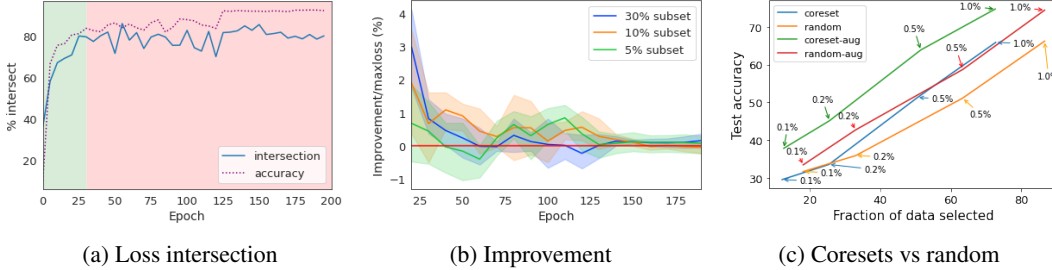

(a) Loss intersection      (b) Improvement      (c) Coresets vs random

Figure 3: Training ResNet20 on full data and augmented coresets extracted from CIFAR10. (a) Intersection between elements of coresets of size 30% and maximum loss subsets of the same size. The intersection increases after the initial phase of training, (b) Accuracy improvement for training on full data and augmented coresets over training on full data and augmented max-loss points. (c) Accuracy vs. fraction of data selected for augmentation during training Resnet20 on CIFAR10.

The pseudo code is given in Alg. 1. The following Lemma upper-bounds the difference between the alignment of the Jacobian and residual for augmented coreset vs. full augmented data.

**Lemma 4.1** *Let $S$ be a coreset that captures the alignment of the full data NTK with residual with an error of at most $\xi$ as in Eq. (4). Augmenting the coreset with perturbations bounded by $\epsilon_0 \leq \frac{1}{n^{\frac{3}{2}}\sqrt{L}}$ captures the alignment of the fully augmented data with the residual by an error of at most*

$$\|\mathcal{J}^T(\boldsymbol{W}^t, \boldsymbol{X}_{aug})\boldsymbol{r} - diag(\boldsymbol{\rho}^t)\mathcal{J}^t(\boldsymbol{W}^t, \boldsymbol{X}_{S^{aug}})\boldsymbol{r}_S\| \leq \xi + \mathcal{O}\left(\sqrt{L}\right). \tag{13}$$

Importantly, since the augmented coresets capture the alignment of the Jacobian of a fully augmented data with residual, they can be trained on in absence of the full data to gain more speedup. This is not the case for existing methods such as max-loss. Besides, the gradient of noisy labeled data points do not cluster in the gradient space (Mirzasoleiman et al., 2020b). Thus, only clean data points are selected for augmentation. This results in an improved performance, as we will show in Sec. 5.1.

### 4.1 CORESET VS. MAX-LOSS STRATEGY FOR DATA AUGMENTATION

In the initial phase of training the NTK goes through rapid changes, which determines the final basin of convergence and network's final performance (Fort et al., 2020). Regularizing deep networks by weight decay or data augmentation mainly affects this initial phase and matters little afterwards (Golatkar et al., 2019). Crucially, augmenting coresets that closely capture the alignment of the NTK with the residual during this initial phase results in a considerable increase in the speed of convergence and generalization performance. On the other hand, augmenting points with maximum loss early in training decreases the alignment between the NTK and the label vector and slows down learning and convergence. Fig. 3b illustrates the superior performance of coreset vs max-loss augmentation. After this initial phase (highlighted in green in Fig. 3a) when the network have a good prediction performance, the gradients for the majority of data point becomes very small. Here, the alignment is mainly captured by the elements with the maximum loss. Hence, as training proceeds, the intersection between the elements of the coresets and examples with maximum loss increases (*c.f.* Fig. 3a).

The following Theorem characterizes the training dynamics of training on the full data and the augmented coresets, using the additive perturbation model in Eq. (1).

**Theorem 4.2** *Let $\mathcal{L}_i$ be $\beta$-smooth, $\mathcal{L}$ be $\lambda$-smooth and satisfy the $\alpha$-PL condition, that is for $\alpha > 0$, $\|\nabla\mathcal{L}(\boldsymbol{W})\|^2 \geq \alpha\mathcal{L}(\boldsymbol{W})$ for all weights $\boldsymbol{W}$. Let $f$ be Lipschitz in $\boldsymbol{X}$ with constant $L'$, and $\bar{L} = \max\{L, L'\}$. Let $G_0$ be the gradient at initializaion, $\sigma_{\max}$ the maximum singular value of the coreset Jacobian at initialization. Choosing $\epsilon_0 \leq \frac{1}{\sigma_{\max}\sqrt{\bar{L}n}}$ and running SGD on full data with augmented coreset using constant step size $\eta = \frac{\alpha}{\lambda\beta}$, we get the following convergence bound:*

$$\mathbb{E}[\|\nabla\mathcal{L}^{f+c_{\text{aug}}}(\boldsymbol{W}^t)\|] \leq \frac{1}{\sqrt{\alpha}}\left(1 - \frac{\alpha\eta}{2}\right)^{\frac{t}{2}}\left(2G_0 + \xi + \mathcal{O}\left(\frac{\sqrt{\bar{L}}}{\sigma_{\max}}\right)\right). \tag{14}$$

Table 1: Accuracy improvement by augmenting subsets found by our method vs. max-loss and random, over improvement of full data augmentation (F.A.) compared to no augmentation (N.A.). The table shows the results for training on CIFAR10 with ResNet20 (C10/R20), SVHN with ResNet32 (SVHN/R32), and CIFAR10-Imbalanced with ResNet32 (C10-IB/R32), with $R = 20$.

| Dataset | N.A. | F.A. | Random | | | Max-loss | | | Ours | | |
|---|---|---|---|---|---|---|---|---|---|---|---|
| | Acc | Acc | 5% | 10% | 30% | 5% | 10% | 30% | 5% | 10% | 30% |
| C10 / R20 | 89.46 | 93.50 | 21.8% | 39.9% | 65.6% | 32.9% | 47.8% | 73.5% | **34.9%** | **51.5%** | **75.0%** |
| C10-IB / R32 | 87.08 | 92.48 | 25.9% | 45.2% | 74.6% | 31.3% | 39.6% | 74.6% | **37.4%** | **49.4%** | 74.8% |
| SVHN / R32 | 95.68 | 97.07 | 5.8% | 36.7% | 64.1% | **35.3%** | **49.7%** | 76.4% | 31.7% | 48.3% | **80.0%** |

The above Theorem shows that training on full data and augmented coresets converges with the same rate as of training on the fully augmented data, to a close neighborhood of the optimal solution. The size of the neighborhood depends on (1) the error of the coreset $\xi$ in Eq. (4), and (2) the error in capturing the alignment of the full augmented data with the residual derived in Lemma 4.1. The first term decrease as the size of the coreset grows, and the second term depends on the network structure.

We also analyze convergence of training only on the augmented coresets, and augmentations modelled as arbitrary linear transformations using a linear model (Wu et al., 2020) in Appendix B.1, B.3.

## 5 EXPERIMENTS

**Setup and baselines.** We evaluate the performance of our approach in two different settings. First, we consider the case where full data is small enough to be trained on. Here, we study the effect of augmenting coresets on the generalization performance of clean and noisy labeled data. Next, we focus on the case where the full training data is too large to be trained on. In this setting, we train only on the augmented coresets. In both cases we use subsets with maximum loss, and random subsets as baselines. For all methods, we select a new augmentation subset every $R$ epochs. We note that while the original method of (Kuchnik & Smith, 2018) selects points based on loss of a fully trained model, to maximize fairness, our max-loss baseline selects a new subset at every subset selection step.

**Data and augmentation.** We apply our method to training ResNet20 and Wide-Resnet-28-10 on CIFAR10, and ResNet32 on CIFAR10-IMB (Long-Tailed CIFAR10 with Imbalance factor of 100 following (Kim & Kim, 2020)) and SVHN datasets. All models are trained for 200 epochs using SGD with 0.9 momentum and learning rate warm-up. We use the strong augmentation proposed by (Wu et al., 2020) to generate 4 distinct augmented examples by randomly sampling 2 augmentations from the same set used by (Cubuk et al., 2019; 2020) to apply to each example. A new set of default augmentations (random crop and horizontal flip) are also applied every $R$ epochs to the original data. We individually measure average time taken for subset selection, gradient descent, and subset augmentation per epoch to compute total training time and speedups. All results are averaged over 5 runs using an Nvidia A40 GPU, and full results are reported in the Appendix.

### 5.1 TRAINING ON FULL DATA AND AUGMENTED SUBSETS

Table 1 demonstrates the accuracy improvement resulted by augmenting subsets of size 5%, 10% and 30% selected by our method vs. max-loss and random over full data augmentation. We observe that augmenting coresets effectively improves generalization, and outperforms augmenting random

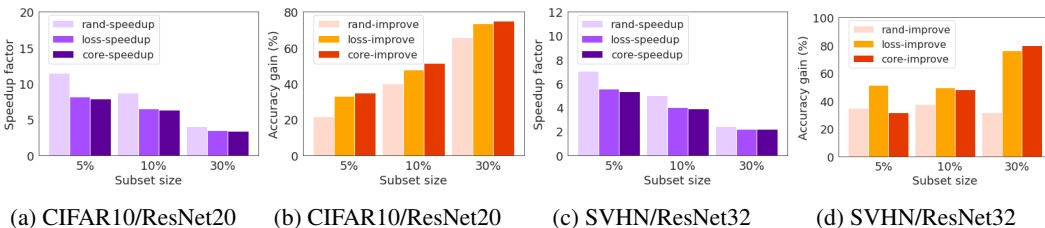

(a) CIFAR10/ResNet20    (b) CIFAR10/ResNet20    (c) SVHN/ResNet32    (d) SVHN/ResNet32

Figure 4: Accuracy improvement and speedups by augmenting subsets found by our method vs. max-loss and random, over improvement of full data augmentation (F.A.) compared to no augmentation (N.A.). The Figure shows the results for CIFAR10 with ResNet20 in terms of (a) speedup and (b) accuracy, and results for SVHN with ResNet32 in terms of (c) speedup and (d) acuracy.

Table 2: Training ResNet20 on CIFAR10 with $50\%$ label noise, with $R = 20$. Accuracy without augmentation is $70.72 \pm 0.20$ and the accuracy of full data augmentation is $75.87 \pm 0.77$. Note that augmenting $50\%$ subsets outperforms augmenting the entire data (marked as **).

| Subset | Random | Loss | Ours |
|--------|--------|------|------|
| 10% | $72.32 \pm 0.14$ | $71.83 \pm 0.13$ | $\mathbf{73.02 \pm 1.06}$ |
| 30% | $74.46 \pm 0.27$ | $72.45 \pm 0.48$ | $\mathbf{74.67 \pm 0.15}$ |
| 50% | $75.36 \pm 0.05$ | $73.23 \pm 0.72$ | $\mathbf{76.20 \pm 0.75}$** |

Table 3: Training ResNet20 and WideResnet-28-10 on CIFAR10 using small subsets, with $R = 1$. Training and augmenting subsets selected by max-loss performed poorly and did not converge.

| Model | Subset | Random | | | Ours | | | Loss |
|-------|--------|--------|------|---------|--------|------|---------|------|
| | | No Aug. | Aug. | Improv. | No Aug. | Aug. | Improv. | Aug. |
| ResNet20 | 0.1% | $31.7 \pm 3.2$ | $33.5 \pm 2.7$ | 5.7% | $29.6 \pm 3.8$ | $\mathbf{37.8 \pm 4.5}$ | $\mathbf{27.7}\%$ | $< 15\%$ |
| | 0.2% | $35.9 \pm 2.1$ | $42.7 \pm 3.9$ | 18.9% | $33.6 \pm 3.2$ | $\mathbf{45.1 \pm 2.3}$ | $\mathbf{34.2}\%$ | $< 15\%$ |
| | 0.5% | $51.1 \pm 2.3$ | $58.7 \pm 1.3$ | $\mathbf{14.9}\%$ | $55.8 \pm 3.1$ | $\mathbf{63.9 \pm 2.1}$ | 14.5% | $< 15\%$ |
| | 1% | $66.2 \pm 1.0$ | $74.4 \pm 0.8$ | 12.4% | $65.9 \pm 4.0$ | $\mathbf{74.7 \pm 1.1}$ | $\mathbf{13.4}\%$ | $< 15\%$ |
| WRN-28-10 | 1% | $61.3 \pm 2.4$ | $57.7 \pm 0.8$ | $-5.9\%$ | $59.9 \pm 2.4$ | $\mathbf{62.1 \pm 3.1}$ | $\mathbf{3.7}\%$ | $< 15\%$ |

and max-loss subsets across different models and datasets. Fig. 5 shows the speedup vs. test accuracy of augmenting subsets of different size selected from CIFAR10 and SVHN by our method vs baselines. We see that our method outperforms the baseline while providing a similar speedup as that of max-loss. On SVHN, we achieve $80.0\%$ of the augmentation accuracy gain by augmenting only $30\%$ size subsets while providing 2.2x speedup, while max-loss only achieves $76.4\%$ with the same speedup. In the Appendix, we also show that our approach is effective when trained with full data even for small augmentation subset sizes of $0.2\%$ and $0.5\%$ selected every epoch.

**Augmenting noisy labeled data.** Table 2 shows the result of augmenting coresets vs. max-loss and random subsets of different sizes selected from CIFAR10 with 50% label noise for training ResNet20. Note that augmenting coresets selected by our method not only outperforms max-loss and random, but provides a superior performance over full data augmentation. This clearly shows the effectiveness of the coresets in capturing the alignment of the NTK with the residual of clean data points.

### 5.2 TRAINING ONLY ON AUGMENTED SUBSETS IN ABSENCE OF FULL DATA

We now evaluate the effectiveness of our approach for training only on the augmented coresets. Our main goal here is to show the accuracy *improvement resulted from augmenting* the coresets. In particular instead of focusing on the quality of the coresets, we aim at showing the effectiveness of the coresets for augmentation. Table 3 shows the test accuracy for training ResNet20 and Wide-ResNet on CIFAR10 when we only train on small augmented coresets of size $0.1\%$ to $1\%$ selected at every epoch ($R = 1$). We see that while the non-augmented coreset performs worse than random in most cases, the augmented coresets outperform augmented random subsets by a large margin. That is, the improvement from augmenting the coresets is significantly larger than random data points. This clearly shows the effectiveness of augmenting the coresets, and the importance of capturing the alignment of the NTK with the residual for data augmentation. Note that training on the augmented max-loss points did not even converge in absence of full data. Finally, Fig. 3c shows the fraction of data selected for augmentation during training of ResNet20 on CIFAR10. We see that the coresets achieve a superior performance by carefully selecting certain data points more often than others.

## 6 CONCLUSION

We showed that data augmentation improves training and generalization by enlarging the smaller singular values of the neural network Jacobian. Then, we proposed a framework to iteratively extract small subsets of training data that when augmented, closely capture the alignment of the fully augmented Jacobian with the label/residual vector. We showed the effectiveness of augmenting the coresets selected by our method to provide a superior generalization performance when added to the full data, in presence of noisy labels, or as a standalone subset.

## 7 REPRODUCIBILITY STATEMENT

Our experimental setup along with training details are introduced in Section 5. Full results containing standard deviations can be found in Appendix C. All code use for our empirical results are also available in the supplementary material, with along with instructions on how to run them.

All proofs for non-trivial theoretical results in the main paper can be found in Appendix A, while additional theoretical results and their respective proofs can be found in Appendix B.

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

# SUPPLEMENTARY MATERIAL:
# DATA-EFFICIENT AUGMENTATION FOR TRAINING NEURAL NETWORKS

## A PROOF OF MAIN RESULTS

### A.1 PROOF FOR LEMMA 3.1

**Proof** Let $\delta_i := \tilde{\sigma}_i - \sigma_i$, where $\mathbb{P}(\delta_i < 0) = p_i$. Assuming uniform probability between $-\|\boldsymbol{E}\|$ to $0$, and between $0$ to $\|\boldsymbol{E}\|$, we have pdf $\rho_i(x)$ for $\delta_i$:

$$\rho_i(x) = \begin{cases} \frac{p_i}{\|\boldsymbol{E}\|}, & \text{if } -\|\boldsymbol{E}\| \leq x < 0 \\ \frac{1-p_i}{\|\boldsymbol{E}\|}, & 0 \leq x \leq \|\boldsymbol{E}\| \\ 0, & \text{otherwise} \end{cases} \tag{15}$$

Taking expectation,

$$\mathbb{E}(\tilde{\sigma}_i - \sigma_i) = \mathbb{E}(\delta_i) = \int_{-\infty}^{\infty} x \rho_i(x) dx \tag{16}$$

$$= \int_{-\|\boldsymbol{E}\|}^{0} x \frac{p_i}{\|\boldsymbol{E}\|} dx + \int_0^{\|\boldsymbol{E}\|} x \frac{1-p_i}{\|\boldsymbol{E}\|} dx \tag{17}$$

$$= -\frac{\|\boldsymbol{E}\| p_i}{2} + \frac{(1-p_i)\|\boldsymbol{E}\|}{2} \tag{18}$$

$$= \frac{(1-2p_i)\|E\|}{2} \tag{19}$$

We also have

$$\mathbb{E}(\delta_i^2) = \int_{-\infty}^{\infty} x^2 \rho_i(x) dx \tag{20}$$

$$= \int_{-\|\boldsymbol{E}\|}^{0} x^2 \frac{p_i}{\|\boldsymbol{E}\|} dx + \int_0^{\|\boldsymbol{E}\|} x^2 \frac{1-p_i}{\|\boldsymbol{E}\|} dx \tag{21}$$

$$= \frac{\|\boldsymbol{E}\|^2 p_i}{3} + \frac{(1-p_i)\|\boldsymbol{E}\|^2}{3} \tag{22}$$

$$= \frac{\|\boldsymbol{E}\|^2}{3} \tag{23}$$

Thus, we have

$$\mathbb{E}(\tilde{\lambda}_i) = \mathbb{E}(\tilde{\sigma}_i^2) \tag{24}$$

$$= \mathbb{E}((\sigma_i + \delta_i)^2) \tag{25}$$

$$= \mathbb{E}(\sigma_i^2 + 2\sigma_i \delta_i + \delta_i^2) \tag{26}$$

$$= \sigma_i^2 + 2\sigma_i \mathbb{E}[\delta_i] + \mathbb{E}[\delta_i^2] \tag{27}$$

$$= \sigma_i^2 + 2\sigma_i \frac{(1-2p_i)\|\boldsymbol{E}\|}{2} + \frac{\|\boldsymbol{E}\|^2}{3} \tag{28}$$

$$= \sigma_i^2 + \sigma_i(1-2p_i)\|\boldsymbol{E}\| + \frac{\|\boldsymbol{E}\|^2}{3} \tag{29}$$

$\blacksquare$

## A.2 PROOF OF THEOREM 3.2

Using Jensen's inequality, we have

$$\mathbb{E}\left[\|\boldsymbol{y} - f(\boldsymbol{X}_{aug}, \boldsymbol{W}^t)\|_2\right] = \mathbb{E}\left[\sqrt{\sum_{i=1}^{n}(1 - \eta\tilde{\lambda}_i)^{2t}(\boldsymbol{u}_i^T\boldsymbol{y})^2} \pm \epsilon\right] \tag{30}$$

$$\leq \sqrt{\mathbb{E}\left[\sum_{i=1}^{n}(1 - \eta\tilde{\lambda}_i)^{2t}(\boldsymbol{u}_i^T\boldsymbol{y})^2\right]} \tag{31}$$

$$\leq \sqrt{\sum_{i=1}^{n}\mathbb{E}\left[(1 - \eta\tilde{\lambda}_i)^{2t}(n)\right]} \tag{32}$$

$$\leq \sqrt{n\sum_{i=1}^{n}(1 - \eta\mathbb{E}\left[\tilde{\lambda}_i\right])^{2t}} \tag{33}$$

$$= \sqrt{n\sum_{i=1}^{n}(1 - \eta\mathbb{E}\left[\tilde{\lambda}_i\right])^{2t}} \tag{34}$$

$$= \sqrt{n\sum_{i=1}^{n}\left(1 - \eta\left(\sigma_i^2 + \sigma_i(1 - 2p_i)\|E\| + \frac{\|E\|^2}{3}\right)\right)^{2t}} \tag{35}$$

## A.3 PROOF OF COROLLARY 3.3

Under the assumptions of Theorem 5.1 of Arora et al. (2019), i.e. where the minimum eigenvalue of the NTK is $\lambda_{\min}(\mathcal{J}\mathcal{J}^T) \geq \lambda_0$ for a constant $\lambda_0 > 0$, and training data $\boldsymbol{X}$ of size $n$ sampled i.i.d. from distribution $D$ and 1-Lipschitz loss $\mathcal{L}$, we have that with probability $\delta/3$, training the over-parameterized neural network with gradient descent for $t \geq \Omega\left(\frac{1}{n\lambda_0}\log\frac{n}{\delta}\right)$ iterations results in the following population loss $\mathcal{L}_D$ (generalization error)

$$\mathcal{L}_D(\boldsymbol{W}^t, \boldsymbol{X}) \leq \sqrt{\frac{2\boldsymbol{y}^T(\mathcal{J}\mathcal{J}^T)^{-1}\boldsymbol{y}}{n}} + \mathcal{O}\left(\frac{\log\frac{n}{\lambda_0\delta}}{n}\right), \tag{36}$$

with high probability of at least $1 - \delta$ over random initialization and training samples.

Hence, using $\lambda_{\min}, \sigma_{\min}$ to denote minimum eigen and singular value respectively of the NTK corresponding to full data, we get

$$\mathcal{L}_{D_{train}}(\boldsymbol{W}^t, \boldsymbol{X}_{train}) \leq \sqrt{\frac{2\frac{1}{\lambda_{\min}}\|y\|^2}{n}} + \mathcal{O}\left(\log\frac{1}{\delta}\right) \tag{37}$$

$$\leq \sqrt{\frac{2}{\sigma_{\min}^2}} + \mathcal{O}\left(\log\frac{1}{\delta}\right). \tag{38}$$

For augmented dataset $\boldsymbol{X}_{aug}$, we have $\tilde{\sigma}_i \leq \sigma_i + \sqrt{n}L\epsilon_0$, hence the improvement in the generalization error is at most

$$\mathcal{L}_{D_{aug}}(\boldsymbol{W}^t, \boldsymbol{X}_{\text{aug}}) \leq \sqrt{\frac{2}{(\sigma_{\min} + \sqrt{n}L\epsilon_0)^2}} + \mathcal{O}\left(\log\frac{1}{\delta}\right). \tag{39}$$

Combining these two results, we obtain Corollary 3.3.

## A.4 PROOF OF LEMMA 4.1

**Proof**

$$\|\mathcal{J}^T(\boldsymbol{W}^t, \boldsymbol{X}_{aug})\boldsymbol{r} - \text{diag}(\boldsymbol{\rho}^t)\mathcal{J}^t(\boldsymbol{W}^t, \boldsymbol{X}_{S^{aug}})\boldsymbol{r}_S\| \tag{40}$$

$$= \|(\mathcal{J}^T(\boldsymbol{W}^t, \boldsymbol{X}) + \boldsymbol{E})\boldsymbol{r} - (\text{diag}(\boldsymbol{\rho}^t)\mathcal{J}^t(\boldsymbol{W}^t, \boldsymbol{X}_S) + \boldsymbol{E}_S)\boldsymbol{r}_S\| \tag{41}$$

$$\leq \|(\mathcal{J}^T(\boldsymbol{W}^t, \boldsymbol{X})\boldsymbol{r} - \text{diag}(\boldsymbol{\rho}^t)\mathcal{J}^t(\boldsymbol{W}^t, \boldsymbol{X}_S)\boldsymbol{r}_S) + \boldsymbol{E}\boldsymbol{r} - \boldsymbol{E}_S\boldsymbol{r}_S\| \tag{42}$$

$$\leq \|(\mathcal{J}^T(\boldsymbol{W}^t, \boldsymbol{X})\boldsymbol{r} - \text{diag}(\boldsymbol{\rho}^t)\mathcal{J}^t(\boldsymbol{W}^t, \boldsymbol{X}_S)\boldsymbol{r}_S)\| + \|\boldsymbol{E}\boldsymbol{r}\| + \|\boldsymbol{E}_S\boldsymbol{r}_S\| \tag{43}$$

Applying definition of coresets, we obtain

$$\|(\mathcal{J}^T(\boldsymbol{W}^t, \boldsymbol{X})\boldsymbol{r} - \text{diag}(\boldsymbol{\rho}^t)\mathcal{J}^t(\boldsymbol{W}^t, \boldsymbol{X}_S)\boldsymbol{r}_S)\| + \|\boldsymbol{E}\boldsymbol{r}\| + \|\boldsymbol{E}_S\boldsymbol{r}_S\| \tag{44}$$

$$\leq \xi + \|\boldsymbol{E}\boldsymbol{r}\| + \|\boldsymbol{E}_S\boldsymbol{r}_S\| \tag{45}$$

$$\leq \xi + 2n^{\frac{3}{2}}L\epsilon_0 \tag{46}$$

$$\leq \xi + 2\sqrt{L} \tag{47}$$

∎

## A.5 PROOF OF THEOREM 4.2

**Proof** In this proof, as shorthand notation, we use $\boldsymbol{X}_f$ and $\boldsymbol{X}_{train}$ interchangeably. We further use $\boldsymbol{X}_c$ to represent the coreset selected from the full data, and $\boldsymbol{X}_{c_{aug}}$ to represent the augmented coreset.

By Theorem 1 of $Bassily\ et\ al.$ (2018), under the $\alpha$-PL assumption for $\mathcal{L}$ and interpolation assumption (i.e. for every sequence $\boldsymbol{W}^1, \boldsymbol{W}^2, \ldots$ such that $\lim_{t \to \infty} \mathcal{L}(\boldsymbol{W}^t, \boldsymbol{X}) = 0$, we have that the loss for each data point $\lim_{t \to \infty} \mathcal{L}(\boldsymbol{W}^t, \boldsymbol{x}_i) = 0$), the convergence of SGD with constant step size is given by

$$\mathbb{E}[\|\nabla\mathcal{L}(\boldsymbol{W}^t, \boldsymbol{X}_{f+c_{\text{aug}}})\|^2] \leq \left(1 - \frac{\alpha\eta}{2}\right)^t \mathcal{L}(\boldsymbol{W}^0, \boldsymbol{X}_{f+c_{\text{aug}}}) \tag{48}$$

$$\leq \frac{1}{\alpha}\left(1 - \frac{\alpha\eta}{2}\right)^t \|\nabla\mathcal{L}(\boldsymbol{W}^0, \boldsymbol{X}_{f+c_{\text{aug}}})\|^2 \tag{49}$$

Using Jensen's inequality, we have

$$\mathbb{E}[\|\nabla\mathcal{L}(\boldsymbol{W}^0, \boldsymbol{X}_{f+c_{\text{aug}}})\|] \tag{50}$$

$$\leq \sqrt{\mathbb{E}[\|\nabla\mathcal{L}(\boldsymbol{W}^t, \boldsymbol{X}_{f+c_{\text{aug}}})\|^2]} \tag{51}$$

$$\leq \frac{1}{\sqrt{\alpha}}\left(1 - \frac{\alpha\eta}{2}\right)^{\frac{t}{2}} \|\nabla\mathcal{L}(\boldsymbol{W}^0, \boldsymbol{X}_{f+c_{\text{aug}}})\| \tag{52}$$

$$\leq \frac{1}{\sqrt{\alpha}}\left(1 - \frac{\alpha\eta}{2}\right)^{\frac{t}{2}} \left(\|\nabla\mathcal{L}(\boldsymbol{W}^0, \boldsymbol{X}_f)\| + \|\nabla\mathcal{L}(\boldsymbol{W}^0, \boldsymbol{X}_{c_{\text{aug}}})\|\right) \tag{53}$$

$$\leq \frac{1}{\sqrt{\alpha}}\left(1 - \frac{\alpha\eta}{2}\right)^{\frac{t}{2}} \left(G_0 + \|(\mathcal{J}(\boldsymbol{W}^0, \boldsymbol{X}_c) + \boldsymbol{E})(\boldsymbol{y} - f(\boldsymbol{W}^0, \boldsymbol{X}_c + \boldsymbol{\epsilon}))\|\right) \tag{54}$$

$$\leq \frac{1}{\sqrt{\alpha}}\left(1 - \frac{\alpha\eta}{2}\right)^{\frac{t}{2}} \tag{55}$$

$$\left(G_0 + \|(\mathcal{J}(\boldsymbol{W}^0, \boldsymbol{X}_c) + \boldsymbol{E})^T(\boldsymbol{y} - f(\boldsymbol{W}^0, \boldsymbol{X}_c) - \nabla_x f(\boldsymbol{W}^0, \boldsymbol{X}_c)^T\boldsymbol{\epsilon} - \mathcal{O}(\boldsymbol{\epsilon}^T\boldsymbol{\epsilon}))\|\right) \tag{56}$$

$$= \frac{1}{\sqrt{\alpha}}\left(1 - \frac{\alpha\eta}{2}\right)^{\frac{t}{2}} \left(G_0 + \|\nabla L(\boldsymbol{W}^0, \boldsymbol{X}_c) - (\mathcal{J}(\boldsymbol{W}^0, \boldsymbol{X}_c)^T(\nabla_x f(\boldsymbol{W}^0, \boldsymbol{X}_c)^T\boldsymbol{\epsilon} + \mathcal{O}(\boldsymbol{\epsilon}^T\boldsymbol{\epsilon})) + \right. \tag{57}$$

$$\boldsymbol{E}(\boldsymbol{y} - f(\boldsymbol{W}^0, \boldsymbol{X}_c + \boldsymbol{\epsilon}))\| \big) \tag{58}$$

$$\leq \frac{1}{\sqrt{\alpha}}\left(1 - \frac{\alpha\eta}{2}\right)^{\frac{t}{2}} \left(G_0 + \|\nabla L(\boldsymbol{W}^0, \boldsymbol{X}_c) - (\mathcal{J}(\boldsymbol{W}^0, \boldsymbol{X}_c)^T(\nabla_x f(\boldsymbol{W}^0, \boldsymbol{X}_c)^T\boldsymbol{\epsilon} + \mathcal{O}(\boldsymbol{\epsilon}^T\boldsymbol{\epsilon}))\| + \right. \tag{59}$$

$$\sqrt{2}\|\boldsymbol{E}\|\big) \tag{60}$$

$$\leq \frac{1}{\sqrt{\alpha}}\left(1 - \frac{\alpha\eta}{2}\right)^{\frac{t}{2}} \left(G_0 + \|\nabla L(\boldsymbol{W}^0, \boldsymbol{X}_c)\| + \sigma_{\max}\bar{L}\sqrt{n}\epsilon_0 + \sigma_{\max}\mathcal{O}(n\epsilon_0^2)) + \sqrt{2n}\bar{L}\epsilon_0\right) \tag{61}$$

$$= \frac{1}{\sqrt{\alpha}}\left(1 - \frac{\alpha\eta}{2}\right)^{\frac{t}{2}} \left(G_0 + \|\nabla L(\boldsymbol{W}^0, \boldsymbol{X}_f)\| + \xi + \sigma_{\max}\bar{L}\sqrt{n}\epsilon_0 + \sigma_{\max}\mathcal{O}(n\epsilon_0^2)) + \sqrt{2n}\bar{L}\epsilon_0\right) \tag{62}$$

$$\leq \frac{1}{\sqrt{\alpha}}\left(1 - \frac{\alpha\eta}{2}\right)^{\frac{t}{2}} \left(2G_0 + \xi + \sigma_{\max}\bar{L}\sqrt{n}\epsilon_0 + \sigma_{\max}\mathcal{O}(n\epsilon_0^2)) + \sqrt{2n}\bar{L}\epsilon_0\right) \tag{63}$$

$\blacksquare$

## A.6 FINDING SUBSETS

Let $S$ be a subset of training data points. Furthermore, assume that there is a mapping $\pi_{w,S} : V \to S$ that for every $\boldsymbol{W}$ assigns every data point $i \in V$ to its closest element $j \in S$, i.e. $j = \pi_{w,S}(i) = \arg\max_{j' \in S} s_{ij'}(\boldsymbol{W})$, where $s_{ij}(\boldsymbol{W}) = C - \|\mathcal{J}^T(\boldsymbol{W}^t, \boldsymbol{x}_i)r_i - \mathcal{J}^T(\boldsymbol{W}^t, \boldsymbol{x}_j)r_j\|$ is the similarity between gradients of $i$ and $j$, and $C \geq \max_{ij} s_{ij}(\boldsymbol{W})$ is a constant. Consider a matrix $\boldsymbol{G}_{\pi_{w,S}} \in \mathbb{R}^{n \times m}$, in which every row $i$ contains gradient of $\pi_w(i)$, i.e. $[\boldsymbol{G}_{\pi_{w,S}}]_{i.} = \mathcal{J}^T(\boldsymbol{W}^t, \boldsymbol{x}_{\pi_{w,S}(i)})r_{\pi_{w,S}(i)}$. The Frobenius norm of the matrix $\boldsymbol{G}_{\pi_w}$ provides an upper-bound on the error of the weighted subset $S$ in capturing the alignment of the residuals of the full training data with the Jacobian matrix. Formally,

$$\|\mathcal{J}^T(\boldsymbol{W}^t, \boldsymbol{X}_{train})\boldsymbol{r}_{train}^t - \boldsymbol{\gamma}_{S^t}\mathcal{J}^T(\boldsymbol{W}^t, [\boldsymbol{X}_{train}]_{.S^t})\boldsymbol{r}_{S^t}\| \leq \|\boldsymbol{G}_{\pi_{w,S}}\|_F, \tag{64}$$

where the weight vector $\boldsymbol{\gamma}_{S^t} \in \mathbb{R}^{|S|}$ contains the number of elements that are mapped to every element $j \in S$ by mapping $\pi_{w,S}$, i.e. $\gamma_j = \sum_{i \in V} \mathbb{1}[\pi_{w,S}(i) = j]$. Hence, the set of training points that closely estimate the projection of the residuals of the full training data on the Jacobian spectrum can be obtained by finding a subset $S$ that minimizes the Frobenius norm of matrix $\boldsymbol{G}_{\pi_{w,S}}$.

## B  ADDITIONAL THEORETICAL RESULTS

### B.1  CONVERGENCE ANALYSIS FOR TRAINING ON THE CORESET AND ITS AUGMENTATION

**Theorem B.1** *Let $\mathcal{L}_i$ be $\beta$-smooth, $\mathcal{L}$ be $\lambda$-smooth and satisfy the $\alpha$-PL condition, that is for $\alpha > 0$, $\|\nabla\mathcal{L}(\boldsymbol{W}, \boldsymbol{X})\|^2 \geq \alpha\mathcal{L}(\boldsymbol{W}, \boldsymbol{X})$ for all weights $\boldsymbol{W}$. Let $\xi$ upper-bound the normed difference in gradients between the weighted coreset and full dataset. Assume that the network $f(\boldsymbol{W}, \boldsymbol{X})$ is Lipschitz in $\boldsymbol{W}$, $\boldsymbol{X}$ with Lipschitz constant $L$ and $L'$ respectively, and $\bar{L} = \max\{L, L'\}$. Let $G_0$ the gradient over the full dataset at initialization, $\sigma_{\max}$ the maximum Jacobian singular value at initialization. Choosing perturbation bound $\epsilon_0 \leq \frac{1}{\sigma_{\max}\sqrt{\bar{L}n}}$ where $\sigma_{\max}$ is the maximum singular value of the coreset Jacobian and $n$ is the size of the original dataset, running SGD on the coreset and its augmentation using constant step size $\eta = \frac{\alpha}{\lambda\beta}$, we get the following convergence bound:*

$$\mathbb{E}[\|\nabla\mathcal{L}(\boldsymbol{W}^t, \boldsymbol{X}_{c+c_{\text{aug}}})|\|] \leq \frac{1}{\sqrt{\alpha}}\left(1 - \frac{\alpha\eta}{2}\right)^{\frac{t}{2}}\left(2G_0 + 2\xi + \mathcal{O}\left(\frac{\bar{L}}{\sigma_{\max}}\right)\right), \tag{65}$$

*where $\boldsymbol{X}_{c+c_{aug}}$ represents the dataset containing the (weighted) coreset and its augmentation.*

**Proof** As in the proof for Theorem 4.2, we begin with the following inequality

$$\mathbb{E}[\|\nabla\mathcal{L}(\boldsymbol{W}^t, \boldsymbol{X}_{c+c_{\text{aug}}})\|^2] \leq \left(1 - \frac{\alpha\eta}{2}\right)^t \mathcal{L}(\boldsymbol{W}^0, \boldsymbol{X}_{c+c_{\text{aug}}}) \tag{66}$$

$$\leq \frac{1}{\alpha}\left(1 - \frac{\alpha\eta}{2}\right)^t \|\nabla\mathcal{L}(\boldsymbol{W}^0, \boldsymbol{X}_{c+c_{\text{aug}}})\|^2 \tag{67}$$

Thus, we can write

$$\mathbb{E}[\|\nabla\mathcal{L}(\boldsymbol{W}^0, \boldsymbol{X}_{c+c_{\text{aug}}})|\|] \tag{68}$$

$$\leq \sqrt{\mathbb{E}[\|\nabla\mathcal{L}(\boldsymbol{W}^t, \boldsymbol{X}_{c+c_{\text{aug}}})\|^2]} \tag{69}$$

$$\leq \frac{1}{\sqrt{\alpha}}\left(1 - \frac{\alpha\eta}{2}\right)^{\frac{t}{2}} \|\nabla\mathcal{L}(\boldsymbol{W}^0, \boldsymbol{X}_{c+c_{\text{aug}}})\| \tag{70}$$

$$\leq \frac{1}{\sqrt{\alpha}}\left(1 - \frac{\alpha\eta}{2}\right)^{\frac{t}{2}} \left(\|\nabla\mathcal{L}(\boldsymbol{W}^0, \boldsymbol{X}_c)\| + \|\nabla\mathcal{L}(\boldsymbol{W}^0, \boldsymbol{X}_{c_{\text{aug}}})\|\right) \tag{71}$$

$$\leq \frac{1}{\sqrt{\alpha}}\left(1 - \frac{\alpha\eta}{2}\right)^{\frac{t}{2}} \left(G_0 + \xi + \|(\mathcal{J}(\boldsymbol{W}^0, \boldsymbol{X}_c) + \boldsymbol{E})(-f(\boldsymbol{W}^0, \boldsymbol{X}_c + \boldsymbol{\epsilon}))\|\right) \tag{72}$$

The rest of the proof is similar to that of Theorem 4.2. ∎

### B.2  LEMMA FOR EIGENVALUES OF CORESET

The following Lemma characterizes the sum of eigenvalues of the NTK associated with the coreset.

**Lemma B.2** *Let $\xi$ be an upper bound of the normed difference in gradient of the weighted coreset and the original dataset, i.e. for full data $\boldsymbol{X}$ and its corresponding coreset $\boldsymbol{X}_S$ with weights $\gamma_S$, and respective residuals $\boldsymbol{r}$, $\boldsymbol{r}_S$, we have the bound $\|\mathcal{J}^T(\boldsymbol{W}^t, \boldsymbol{X})\boldsymbol{r}^t - \gamma_S\mathcal{J}^T(\boldsymbol{W}^t, \boldsymbol{X}_S)\boldsymbol{r}_S^t\| \leq \xi$. Let $\{\lambda_i\}_{i=1}^k$ be the eigenvalues of the NTK associated with the coreset. Then we have that*

$$\sqrt{\sum_{i=1}^k \lambda_i} \geq \frac{|\|\mathcal{J}^T(\boldsymbol{W}^t, \boldsymbol{X})\boldsymbol{r}^t\| - \xi|}{\|\boldsymbol{r}_S^t\|}.$$

**Proof** Let singular values of coreset Jacobian be $\sigma_i$. Let $\mathcal{J}^T(\boldsymbol{W}^t, \boldsymbol{X})\boldsymbol{r}^t = \gamma_S\mathcal{J}^T(\boldsymbol{W}^t, \boldsymbol{X}_S)\boldsymbol{r}_S^t + \xi_S$ where $\|\xi_S\| \leq \xi$.

Taking Frobenius norm, we get

$$\|\gamma_S \mathcal{J}^T(\boldsymbol{W}^t, \boldsymbol{X}_S)\boldsymbol{r}_S^t\| = \|\mathcal{J}^T(\boldsymbol{W}^t, \boldsymbol{X})\boldsymbol{r}^t - \xi_S\| \tag{73}$$

$$\Rightarrow \|\gamma_S \mathcal{J}^T(\boldsymbol{W}^t, \boldsymbol{X}_S)\|\|\boldsymbol{r}_S^t\| \geq \|\mathcal{J}^T(\boldsymbol{W}^t, \boldsymbol{X})\boldsymbol{r}^t - \xi_S\| \tag{74}$$

$$\Rightarrow \|\gamma_S \mathcal{J}^T(\boldsymbol{W}^t, \boldsymbol{X}_S)\| \geq \frac{\|\mathcal{J}^T(\boldsymbol{W}^t, \boldsymbol{X})\boldsymbol{r}^t - \xi_S\|}{\|\boldsymbol{r}_S^t\|} \tag{75}$$

$$\Rightarrow \sqrt{\sum_{i=1}^s \sigma_i^2} \geq \frac{\|\mathcal{J}^T(\boldsymbol{W}^t, \boldsymbol{X})\boldsymbol{r}^t - \xi_S\|}{\|\boldsymbol{r}_S^t\|} \tag{76}$$

$$\Rightarrow \sqrt{\sum_{i=1}^s \lambda_i} \geq \frac{\|\mathcal{J}^T(\boldsymbol{W}^t, \boldsymbol{X})\boldsymbol{r}^t - \xi_S\|}{\|\boldsymbol{r}_S^t\|} \tag{77}$$

$$\Rightarrow \sqrt{\sum_{i=1}^s \lambda_i} \geq \frac{|\|\mathcal{J}^T(\boldsymbol{W}^t, \boldsymbol{X})\boldsymbol{r}^t\| - \xi|}{\|\boldsymbol{r}_S^t\|} \quad \text{by reverse triangle inequality} \tag{78}$$

$\blacksquare$

We can make the following observations: For overparameterized networks, with bounded activation functions and labels, e.g. softmax and one-hot encoding, the norm of the residual vector is bounded, and the gradient norm is likely to be much larger than residual, especially when dimension of gradient is large. In this case, the Jacobian matrix associated with small weighted coresets found by solving Eq. (11), have large singular values.

### B.3 Augmentation as Linear Transformation: Linear Model Analysis

We introduce a simplified linear model to extend our theoretical analysis to augmentations modelled as linear transformation matrices $F$ applied to the original training data. These augmentations are also originally studied by Wu et al. (2020). In this section, we specifically study the effect of these augmentations using a linear model when applied to coresets.

### B.4 Linear case (1 input layer, 1 output layer)

**Lemma B.3 (Augmented coreset gradient bounds: Linear)** *Let $f$ be a simple linear model with weights $\boldsymbol{W} \in \mathbb{R}^{d \times C}$ where $f(\boldsymbol{W}, \boldsymbol{x}_i) = \boldsymbol{W}^T \boldsymbol{x}_i$, trained on mean squared loss function $\mathcal{L}$. Let $F \in \mathbb{R}^{d \times d}$ be a common linear augmentation matrix with norm $\|F\|$ with augmentation $\boldsymbol{x}_i^{\mathrm{aug}}$ given by $F\boldsymbol{x}_i$. Let coreset be of size $k$ and full dataset be of size $n$. Further assume that the predicted label of $\boldsymbol{x}_i$ and its augmentation $F\boldsymbol{x}_i$ are sufficiently close, i.e. there exists $\omega$ such that $\boldsymbol{W}^T(F\boldsymbol{x}_i) = \boldsymbol{W}^T \boldsymbol{x}_i + z_i$, $\|z_i\| \leq \omega \, \forall i$. Let $\xi$ upper-bound the normed difference in gradients between the weighted coreset and full dataset. Then, the normed difference between the gradient of the augmented full data and augmented coreset is given by*

$$\|\sum_{i \in V} \nabla \mathcal{L}(\boldsymbol{W}, \boldsymbol{x}_i^{\mathrm{aug}}) - \sum_{j=1}^k \gamma_{s_j} \nabla \mathcal{L}(\boldsymbol{W}, \boldsymbol{x}_{s_j}^{\mathrm{aug}})\| \leq \|F\|(\xi + \sqrt{d}n\omega)$$

*for some (small) constant $\xi$.*

**Proof** By our assumption, we can begin with,

$$\|\sum_{i \in V} \nabla \mathcal{L}(\boldsymbol{W}, \boldsymbol{x}_i) - \sum_{j=1}^k \gamma_{s_j} \nabla \mathcal{L}(\boldsymbol{W}, \boldsymbol{x}_{s_j})\| \leq \xi \tag{79}$$

Furthermore, by Mirzasoleiman et al. (2020a), we know that sum of the coreset weights $\gamma_{s_j}$ is given by $\sum_{j=1}^{k=1} \gamma_{s_j} \leq n$.

Hence,

$$\|\sum_{i \in V} \nabla \mathcal{L}(\boldsymbol{W}, \boldsymbol{x}_i^{\mathrm{aug}}) - \sum_{j=1}^{k} \gamma_{s_j} \nabla \mathcal{L}(\boldsymbol{W}, \boldsymbol{x}_{s_j}^{\mathrm{aug}})\| \tag{80}$$

$$= \|\sum_{i \in V} (\mathcal{J}(\boldsymbol{W}, \boldsymbol{x}_i^{\mathrm{aug}}))^T [\boldsymbol{W}^T (F\boldsymbol{x}_i) - y_i] - \sum_{j=1}^{k} \gamma_{s_j} (\mathcal{J}(\boldsymbol{W}, \boldsymbol{x}_{s_j}^{\mathrm{aug}}))^T [\boldsymbol{W}^T (F\boldsymbol{x}_{s_j}) - y_{s_j}]\| \tag{81}$$

$$= \|\sum_{i \in V} F\boldsymbol{x}_i [\boldsymbol{W}^T (F\boldsymbol{x}_i) - y_i] - \sum_{j=1}^{k} \gamma_{s_j} F\boldsymbol{x}_{s_j} [\boldsymbol{W}^T (F\boldsymbol{x}_{s_j}) - y_{s_j}]\| \tag{82}$$

$$= \|F \sum_{i \in V} \boldsymbol{x}_i (\boldsymbol{W}^T \boldsymbol{x}_i - y_i) - F \sum_{j=1}^{k} \gamma_{s_j} \boldsymbol{x}_{s_j} (\boldsymbol{W}^T \boldsymbol{x}_{s_j} + z_i - y_{s_j})\| \tag{83}$$

$$= \|F \sum_{i \in V} \nabla L(\boldsymbol{W}, \boldsymbol{x}_i) - F \sum_{j=1}^{k} \gamma_{s_j} \nabla L(\boldsymbol{W}, \boldsymbol{x}_{s_j}) - F \sum_{j=1}^{k} \gamma_{s_j} \boldsymbol{x}_{s_j} z_{s_j}\| \tag{84}$$

$$\leq \|F\| \|\sum_{i \in V} \nabla L(\boldsymbol{W}, \boldsymbol{x}_i) - \sum_{j=1}^{k} \gamma_{s_j} \nabla L(\boldsymbol{W}, \boldsymbol{x}_{s_j})\| + \|F\| \|\sum_{j=1}^{k} \gamma_{s_j} \boldsymbol{x}_{s_j} z_{s_j}\| \tag{85}$$

$$\leq \|F\|\xi + \sqrt{d}\|F\|n\omega \tag{86}$$
$$= \|F\|(\xi + \sqrt{d}n\omega) \tag{87}$$

∎

**Corollary B.4** *In the simplified linear case above, the difference in gradients of the full training data with its augmentations ($\nabla \mathcal{L}(\boldsymbol{W}, \boldsymbol{X}_{f+aug})$) and gradients of the coreset with its augmentations ($\nabla \mathcal{L}(\boldsymbol{W}, \boldsymbol{X}_{c+c_{aug}})$) can be bounded by*

$$\|\nabla \mathcal{L}(\boldsymbol{W}, \boldsymbol{X}_{f+aug}) - \nabla \mathcal{L}(\boldsymbol{W}, \boldsymbol{X}_{c+c_{aug}})\| \leq (\|F\|+1)\xi + \sqrt{d}\|F\|n\omega$$

**Proof** Applying Eq. (79) and Lemma B.3, we obtain

$$\|\nabla \mathcal{L}(\boldsymbol{W}, \boldsymbol{X}_{f+aug}) - \nabla \mathcal{L}(\boldsymbol{W}, \boldsymbol{X}_{c+c_{aug}})\| \tag{88}$$
$$= \|(\nabla \mathcal{L}(\boldsymbol{W}, \boldsymbol{X}_f) + \nabla \mathcal{L}(\boldsymbol{W}, \boldsymbol{X}_{aug})) - (\nabla \mathcal{L}(\boldsymbol{W}, \boldsymbol{X}_c) + \nabla \mathcal{L}(\boldsymbol{W}, \boldsymbol{X}_{c_{aug}}))\| \tag{89}$$
$$= \|(\nabla \mathcal{L}(\boldsymbol{W}, \boldsymbol{X}_f) - \nabla \mathcal{L}(\boldsymbol{W}, \boldsymbol{X}_c)) + (\nabla \mathcal{L}(\boldsymbol{W}, \boldsymbol{X}_{aug}) - \nabla \mathcal{L}(\boldsymbol{W}, \boldsymbol{X}_{c_{aug}}))\| \tag{90}$$
$$\leq \|(\nabla \mathcal{L}(\boldsymbol{W}, \boldsymbol{X}_f) - \nabla \mathcal{L}(\boldsymbol{W}, \boldsymbol{X}_c))\| + \|(\nabla \mathcal{L}(\boldsymbol{W}, \boldsymbol{X}_{aug}) - \nabla \mathcal{L}(\boldsymbol{W}, \boldsymbol{X}_{c_{aug}}))\| \tag{91}$$
$$\leq \xi + \|F\|(\xi + \sqrt{d}n\omega) \tag{92}$$
$$= (\|F\|+1)\xi + \sqrt{d}\|F\|n\omega \tag{93}$$

∎

**Theorem B.5 (Convergence of linear model)** *Let $f$ be a linear model with weights $\boldsymbol{W}$ and augmentation be represented by the common linear transformation $F$. Let $\mathcal{L}_i$ be $\beta$-smooth, $\mathcal{L}$ be $\lambda$-smooth and satisfy the $\alpha$-PL condition, that is for $\alpha > 0$, $\|\nabla \mathcal{L}(\boldsymbol{W}, \boldsymbol{X})\|^2 \geq \alpha \mathcal{L}(\boldsymbol{W}, \boldsymbol{X})$ for all weights $\boldsymbol{W}$. Let $\xi$ upper-bound the normed difference in gradients between the weighted coreset and full dataset and $\omega$ bound $\boldsymbol{W}^T(F\boldsymbol{x}_i) = \boldsymbol{W}^T \boldsymbol{x}_i + z_i$, $\|z_i\| \leq \omega \ \forall i$. Let $G_0'$ be the gradient over the full dataset and its augmentations at initialization. Then, running SGD on the size $k$ coreset with its augmentation using constant step size $\eta = \frac{\alpha}{\lambda\beta}$, we get the following convergence bound:*

$$\mathbb{E}[\|\nabla \mathcal{L}(\boldsymbol{W}^t, \boldsymbol{X}_{c+c_{\mathrm{aug}}})\|] \leq \frac{1}{\sqrt{\alpha}} \left(1 - \frac{\alpha\eta}{2}\right)^{\frac{t}{2}} \left(G_0' + (\|F\|+1)\xi + \sqrt{d}\|F\|n\omega\right)$$

**Proof** From $Bassily\,et\,al.$ (2018), we have

$$\mathbb{E}[\|\nabla\mathcal{L}(\boldsymbol{W}^t, \boldsymbol{X}_{c+c_{\mathrm{aug}}})\|^2] \leq \left(1 - \frac{\alpha\eta}{2}\right)^t \mathcal{L}(\boldsymbol{W}^0, \boldsymbol{X}_{c+c_{\mathrm{aug}}}) \tag{94}$$

$$\leq \frac{1}{\alpha}\left(1 - \frac{\alpha\eta}{2}\right)^t \|\nabla\mathcal{L}(\boldsymbol{W}^0, \boldsymbol{X}_{c+c_{\mathrm{aug}}})\|^2 \tag{95}$$

$$\tag{96}$$

Using Jensen's inequality, we have

$$\mathbb{E}[\|\nabla\mathcal{L}(\boldsymbol{W}^t, \boldsymbol{X}_{c+c_{\mathrm{aug}}})\|] \tag{97}$$

$$\leq \sqrt{\mathbb{E}[\|\nabla\mathcal{L}(\boldsymbol{W}^t, \boldsymbol{X}_{c+c_{\mathrm{aug}}})\|^2]} \tag{98}$$

$$\leq \frac{1}{\sqrt{\alpha}}\left(1 - \frac{\alpha\eta}{2}\right)^{\frac{t}{2}} \|\nabla\mathcal{L}(\boldsymbol{W}^0, \boldsymbol{X}_{c+c_{\mathrm{aug}}})\| \tag{99}$$

$$\leq \frac{1}{\sqrt{\alpha}}\left(1 - \frac{\alpha\eta}{2}\right)^{\frac{t}{2}} \left(G'_0 + (\|F\|+1)\xi + \sqrt{d}\|F\|n\omega\right) \tag{100}$$

where the last inequality follows from applying Corollary B.4. ∎

## C  APPENDIX C: EXTRA EXPERIMENTS

### C.1  FULL RESULTS FOR TABLE 1

This section contains full experiment results including standard deviations and the full augmentation benchmark for Table 1. Augmenting coresets of size 10% achieves 51% of the improvement obtained from augmentation of the full data and further enjoys a 6x speedup in training time on CIFAR10. This speedup becomes more significant when using strong augmentation techniques which are mostly computationally demanding, especially when applied to the entire dataset.

Table 4: Supplementary table for Table 1 - Test accuracy on CIFAR10 + ResNet20, SVHN + ResNet32, CIFAR10-Imbalanced + ResNet32 including standard deviation errors and full dataset augmentation accuracy.

| Method | Size | CIFAR10 | CIFAR10-IMB | SVHN |
|---|---|---|---|---|
| None | 0% | $89.46 \pm 0.17\%$ | $87.08 \pm 0.50\%$ | $95.676 \pm 0.108\%$ |
| Random | 5% | $90.34 \pm 0.18\%$ | $88.48 \pm 0.25\%$ | $95.760 \pm 0.084\%$ |
| | 10% | $91.07 \pm 0.13\%$ | $89.52 \pm 0.15\%$ | $96.187 \pm 0.112\%$ |
| | 30% | $92.11 \pm 0.12\%$ | $91.11 \pm 0.18\%$ | $96.569 \pm 0.073\%$ |
| Max-Loss | 5% | $90.79 \pm 0.19\%$ | $88.77 \pm 0.35\%$ | $\mathbf{96.165 \pm 0.108}\%$ |
| | 10% | $91.39 \pm 0.08\%$ | $89.22 \pm 0.48\%$ | $\mathbf{96.370 \pm 0.076}\%$ |
| | 30% | $92.43 \pm 0.07\%$ | $91.11 \pm 0.25\%$ | $96.735 \pm 0.068\%$ |
| Coreset | 5% | $\mathbf{90.87 \pm 0.05}\%$ | $\mathbf{89.10 \pm 0.41}\%$ | $96.121 \pm 0.055\%$ |
| | 10% | $\mathbf{91.54 \pm 0.19}\%$ | $\mathbf{89.75 \pm 0.52}\%$ | $96.354 \pm 0.091\%$ |
| | 30% | $\mathbf{92.49 \pm 0.15}\%$ | $\mathbf{91.12 \pm 0.26}\%$ | $\mathbf{96.791 \pm 0.051}\%$ |
| All | 100% | $93.50 \pm 0.25\%$ | $92.48 \pm 0.34\%$ | $97.068 \pm 0.030\%$ |

### C.2  SPEED-UP MEASUREMENTS

We measure the improvement in training time in the case of training on full data and augmenting subsets of various sizes. While our method yields similar or slightly lower speed-up to the max-loss policy and random approach respectively, our resulting accuracy outperforms these two approaches. For example, for SVHN/Resnet32 using 30% coresets, we sacrifice 10% of the speed-up to obtain an additional 24.8% of the gain in accuracy from full data augmentation when compared to a random subset of the same size.

Table 5: Speedup on CIFAR10 + ResNet20 (C10/R20), SVHN + ResNet32 (SVHN/R32).

| Dataset | Full Aug. | Ours | | | | | | Max loss. | Random. |
|---|---|---|---|---|---|---|---|---|---|
| | 100% | 5% | 10% | 15% | 20% | 25% | 30% | 30% | 30% |
| C10 / R20 | 1x | 7.93x | 6.31x | 4.46x | 4.27x | 3.41x | 3.43x | 3.48x | 4.03x |
| SVHN / R32 | 1x | 5.35x | 3.93x | 3.40x | 2.80x | 2.49x | 2.18x | 2.21x | 2.43x |

## C.3 TRAINING ON FULL DATA AND AUGMENTING SMALL SUBSETS RE-SELECTED EVERY EPOCH

We apply our proposed method to select a new subset for augmentation every epoch (i.e. using $R = 1$) and compare our results with other approaches using accuracy and percentage of data not selected (NS). We see that while the max-loss policy selects a small fraction of data points over and over and random uniformly selects all the data points, our approach successfully finds the smallest subset of data points that are the most crucial for data augmentation. Hence, it can achieve a superior accuracy than max-loss policy, while augmenting only slightly more examples. This confirms the data-efficiency of our approach. This is especially evident when using coresets of size 0.2%. Furthermore, despite the random baseline using a significantly larger percentage of data, it is outperformed by our approach in both data-efficiency and accuracy. We emphasize that results in this table is different from that of Table 4, as default augmentations on the full training data are performed once every $R = 1$ epochs instead of every $R = 20$ epochs. Since selecting subsets at every epoch can be computationally expensive, we only perform these experiments on small coresets and hence still enjoy good speedups compared to full data augmentation. This shows that our approach is still effective at very small subset sizes, hence can be computationally efficient even when subsets are re-selected every epoch.

Table 6: Training on full data and selecting a new subset for augmentation every epoch ($R = 1$).

| Subset | Random | | Max-loss Policy | | Ours | |
|---|---|---|---|---|---|---|
| | Acc | NS (%) | Acc | NS (%) | Acc | NS (%) |
| 0% | $91.96 \pm 0.12$ | – | $91.96 \pm 0.12$ | – | $91.96 \pm 0.12$ | – |
| 0.2% | $92.22 \pm 0.22$ | $67.03 \pm 0.04$ | $91.94 \pm 0.12$ | $86.70 \pm 0.15$ | $\mathbf{92.26} \pm 0.13$ | $79.19 \pm 1.10$ |
| 0.5% | $92.06 \pm 0.17$ | $36.70 \pm 0.18$ | $92.20 \pm 0.13$ | $76.80 \pm 0.31$ | $\mathbf{92.27} \pm 0.08$ | $63.23 \pm 0.35$ |

## C.4 ADDITIONAL VISUALIZATIONS FOR TRAINING ON CORESETS AND ITS AUGMENTATIONS - MEASURING TRAINING DYNAMICS OVER TIME

We include additional visualizations in Figure 5 for training on coresets and its augmentations as supplementary plots to Figure 3c and Table 3. We plot metrics obtained during each point (epoch) of the training process based on percentage of data selected/used and test accuracy achieved. All metrics are averaged over 5 runs and obtained using $R = 1$. These plots demonstrate that coreset augmentation approaches outperform random augmentation baselines throughout the training process. Furthermore, they show that augmentation of coresets result in a larger increase in test accuracy compared to augmentation of randomly selected training examples, especially for small subset sizes.

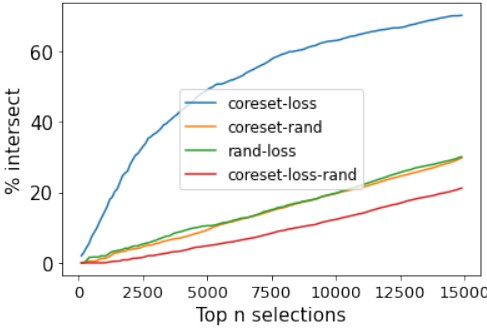

Figure 6: Intersection between max-loss and coresets in the top $N$ points selected aggregated across the entire training process. Here, we show the increasing overlap between max-loss and coreset points as $N$ grows.

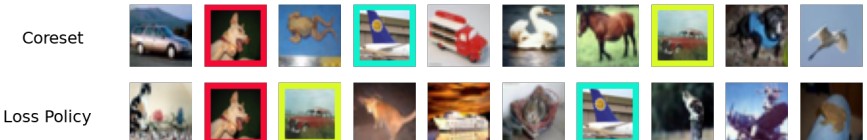

Figure 7: Qualitative evaluation of coreset and max-loss points.

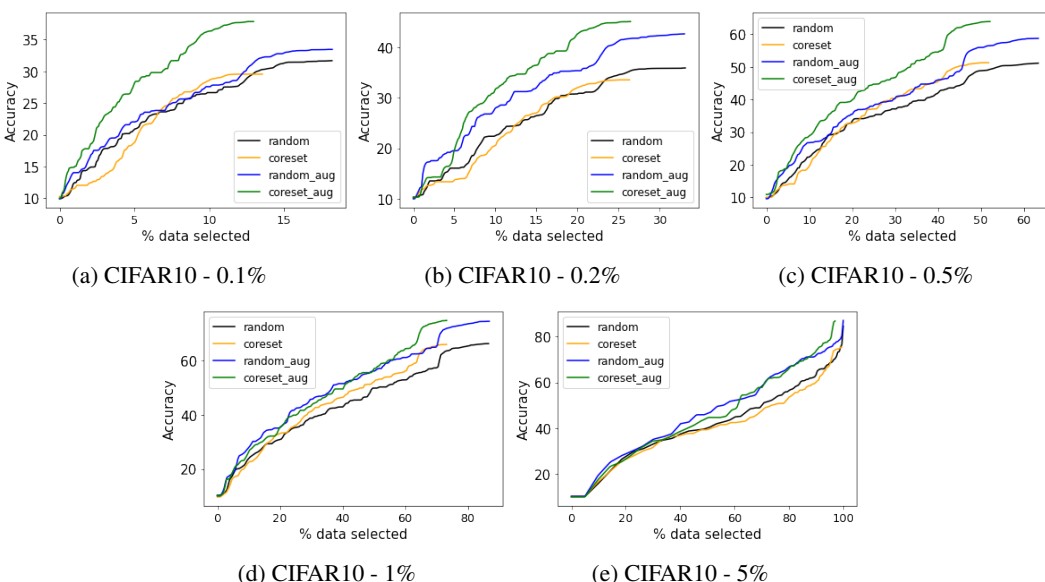

Figure 5: Supplementary plots for Figure 3c: Training on coreset and its augmentation compared to random baseline, measured using test accuracy against percentage of data used on CIFAR10 dataset across various subset sizes. Accuracy and percentage of data used are measured at every epoch and averaged over 5 runs.

## C.5  INTERSECTION OF MAX-LOSS POLICY AND CORESETS

Figure $3a$ depicts the increase in intersection between max-loss subsets and coresets over time. In addition, we also aggregate $30\%$ subsets selected every $R = 20$ epochs using both approaches over the entire training process to compute intersection between the top $N$ selected data points. Our plots in Figure 6 suggest that a similar pattern holds in this setting. We also qualitatively visualize this in Figure 7.

