# OpenReview forum: "Data-Efficient Augmentation for Training Neural Networks"
_ICLR.cc/2022/Conference — ICLR 2022 Submitted_

### Official Review · Reviewer_Q8Ag · 2021-11-02

**Correctness:** 3
**Technical Novelty And Significance:** 3
**Empirical Novelty And Significance:** 3
**Recommendation:** 3
**Confidence:** 3

**Main Review:**

Strengths:
1) The motivation of the paper is clearly justified.
2) The presentation of the theoretical analysis and the algorithm are clear and sound.
3) The results shows improvement over the max-loss method on some datasets.

Weakness:
1) What is the extra computation cost for finding the coreset compared with that of  the random and max-loss baseline?
2) Does the coreset change significantly at different epochs?
3) It would be interesting to show the performance when transferring the core-set found on one architecture (e.g., ResNet20) to train on a different architecture (e.g., Wid-ResNet). Will transfer the corset leads to performance loss?
4) You might referred to the wrong figure in the sentence “Figure 3b depicts the increase in intersection between max-loss subsets and coresets over time”
5) The paper only did experiments on small datasets other than large datasets such as ImageNet.



**Summary Of The Paper:**

This paper shows that data augmentation can speed up learning by enlarging the smaller singular values of the Jacobian. Following this idea, this paper proposed a framework to iteratively extract small subsets of training data that captures the alignment of NTK with the residual when augmented.


**Summary Of The Review:**

This paper proposed a data efficient augmentation framework with extensive theoretical analysis. The presentation and the proposed method are sound. However, the lack of experimental results on large datasets such as ImageNet makes the results of the paper much less convincing.

---

> ### Author Response · Authors · 2021-11-12
> **Reply to Reviewer Q8Ag**
>
>
> Thank you for the feedback. We clarify some of the reviewer’s concerns below
> Speedup: Figure 4 and Table 5 shows the full speedup of coreset methods compared to max-loss and random baselines. To calculate the running time, as we discussed in Sec 5 under “Data and Augmentation”, we measured the average time taken for subset selection, gradient descent, and subset augmentation to compute total training time and speedups. For example in Sec 5.1, the speedup is calculated as the ratio of the total running time for training on the full data and augmented subsets over the total running time for training on the full data and its augmentations. We will further clarify this in the manuscript. We note that computing coresets is extremely fast, as it does not involve any gradient calculation and instead only requires a single forward pass through the network. As such, our coreset approach rivals the speed-up provided by max-loss.
>
> Coresets over time: Figure 3(a) shows an increasing overlap between coreset points and max-loss points as training progresses. As mentioned in the introduction: “We iteratively find small weighted subsets (coresets) that when augmented, closely capture the alignment of the full augmented data with the label/residual, at every point during the training. Our key observation is that early in training, this alignment is best captured by data points that are highly representative of their classes. However towards the end of training when the network converges, data points with maximum loss best capture this alignment.”
>
> Coreset transferability: We do agree that the transferability of coresets across different model architectures will be an extremely interesting topic to explore (i.e. seeing whether coresets are model-independent). However, we note that extracting the coresets is fast, hence they can be easily found for another architecture directly during the training.
> We will leave this to future work.
>
> We thank the reviewer for pointing out the incorrect figure reference and will correct this. We also agree that experiments on larger datasets will be insightful, and hence we are also currently running our experiments on Places Dataset and ImageNet with ResNet56. If these complete in time, the results will be included in our revision. We note that the experiments conducted on WideResNet-28-10 in Table 3 using only 1% subset sizes on CIFAR10 takes over a day to train on an A40 GPU. Running each experiment 5 times to obtain reliable results makes them even more computationally expensive, as such we initially chose to omit experiments on larger datasets like ImageNet, and instead increased complexity through noise and imbalance instead.

---

> > ### Comment · Reviewer_Q8Ag · 2021-11-29
> > **Thanks the authors for the response**
> >
> > Thanks the authors for the response. I have read the authors' response as well as the comments from other reviewers. My primary concern is the motivation of this work which is still not resolved: the key contribution of the work is to use a subset of data for augmentation. When the dataset itself is small, the benefit of selecting a subset is not that significant. It would be more convincing if the authors could conduct their experiments on large datasets such as ImageNet, which will make the work much stronger and more motivated.

---

### Official Review · Reviewer_1cJe · 2021-11-02

**Correctness:** 3
**Technical Novelty And Significance:** 4
**Empirical Novelty And Significance:** 3
**Recommendation:** 6
**Confidence:** 3

**Main Review:**

**Strengths**

I appreciate that this paper provides rigorous theoretical analysis to data augmentation, its connection to NTK and effect on eigenvalues and how data augmentation affects the training dynamics and generalization, and it also provides mathematical proofs to the theorems proposed.

In the presence of noisy labels, the proposed technique is outperforming reference methods.

And even when the selected subset is very small compared to randomly augmenting datasets of similar size the proposed method is capable of providing good augmentation resulting and a decent increase in performance.

**Weaknesses**

As I understand one of the observations in section 3.1 suggests that augmentation cannot change the singular values considerably for diverse datasets, but we still see a gain in performance when augmenting these datasets.
My concern is if we have a more diverse dataset than the ones experimented on, according to my understanding we will have a larger number of singular values and that will limit the ability to find a subset to augment and we need to increase the size of the subset which can largely limit the effectiveness of the method.

In my opinion, one of the main strengths of the method is when we have a very large dataset but unfortunately, the authors didn’t provide any experiments regarding large datasets like ImageNet.

The proposed method outperforms the baseline when the size of the subset is fairly small, but when increasing the size into reasonable percentages the improvement is marginal see Table 4 in appendix C.

**Suggestion To Authors**

Though the whole paper is nicely executed and demonstrated, in my opinion, some of the figures aren’t easy to read e.g. (Figure 4) I suggest splitting the figure into two side by side showing the speed gain and accuracy improvement separately.


**Summary Of The Paper:**

The authors model data augmentation as an additive perturbation and analyze its effect on training dynamics and how it enlarges the smaller singular values of the network jacobian.
Then they propose a new method to iteratively extract a subset of the training data that when augmented closely capture the full augmented data dynamics.
Authors show that by augmenting this subset combined with full training data they can outperform the state-of-the-art method by 7.7% on CIFAR-10 and 4.7% on SVHN while achieving 6.3x and 2.2x speedup respectively.

**Summary Of The Review:**

I think the paper is of interest, especially the theoretical analysis. It is well written and provides rigorous mathematical analysis to data augmentation and proofs for all the theorems, lemmas and corollaries mentioned.

But I believe the main application of the proposed method is in large scale dataset settings, while in small or medium-size datasets augmenting the full dataset is not an issue, but that wasn’t presented (see my concerns in the previous section).
Also, the improvement when increasing subsampling size is marginal, especially with diverse datasets.
This problem and the lack of experiments with large and diverse datasets prevents me from giving a higher recommendation of acceptance.

---

> ### Author Response · Authors · 2021-11-12
> **Reply to Reviewer 1cJe**
>
> We thank the reviewer for acknowledging our rigorous theoretical analysis of data augmentation, and how it affects the training dynamics and generalization.
>
> We reported our experimental results on CIFAR10 (as well as imbalanced and noisy CIFAR10) as strong data augmentation is indeed very expensive. For example, only 1 run of training CIFAR10 on ResNet20 with strong full data augmentation takes ~16 hours on an A40 GPU. For training WideResNet-28-10 on CIFAR10 in Table 3, using only 1% subset sizes with strong augmentation takes over a day to train once on an A40 GPU. To address the reviewer’s comment on large scale experiments, we are running our experiments on larger datasets such as Places dataset and TinyImageNet with ResNet56, and we will report the results if they complete in time.
>
> Regarding diverse datasets, the significant gains from augmentation despite no change in the overall singular value distribution actually corroborates our theory in Eq. (6). As we explained after Eq. (6), for the singular values that do not change, the speedup along the corresponding singular direction is equal to $||E||^2/3$.
>
> Regarding the size of the subset, indeed smaller subsets are required if the data has some underlying (clusterable) structure. This is often the case for real-world datasets. However, if the data points do not have any underlying structure, all the strategies, including our method, random, and max loss, need to select almost all the data points. With a similar reasoning, for larger subsets, coresets, random, and max-loss perform similarly as they all can capture the underlying structure with a lot of data points. Considering the substantial costs of strong data augmentation reported above, we believe that augmenting small subsets is the most beneficial and important scenario for training on large datasets.
>
> We will also modify our figures accordingly as suggested by the reviewer. Thank you!

---

> > ### Comment · Reviewer_1cJe · 2021-11-29
> > **Response to Authors**
> >
> > Thanks to the authors for their response, and I appreciate their explanation!
> >
> > I have read the authors' responses as well as the comments from other reviewers. But the main limitation remains as there are no experiments conducted on large scale datasets like ImageNet. Although understandably, training on such a large dataset can be very expensive, it would have been more convincing. Still training on middle size datasets like TinyImageNet will also be better than focusing on CIFAR10.
> > As this problem remains I am keeping my review the same.

---

### Official Review · Reviewer_fQLH · 2021-11-03

**Correctness:** 3
**Technical Novelty And Significance:** 2
**Empirical Novelty And Significance:** 2
**Recommendation:** 3
**Confidence:** 4

**Main Review:**

The main contribution of this paper is, in my opinion, a sound theoretical analysis of the impact of input perturbations on the singular values of the Jacobian, or the eigenvalues of the Neural Tangent Kernel (NTK), that may inspire other researchers. However, the limitations of the analysis with respect to real-world image data augmentation, as well as the limited motivation, in my opinion, for the need for the proposed method, have an overall negative impact in my assessment of the significance of this work. I elaborate on these concerns below.

First, the whole theoretical analysis, built upon that of Rajput et al. (2019), is based on modelling data augmentation as additive perturbations of the inputs. This is stated at the beginning of Section 2 and never recalled as a limitation again in the paper (except for one section in the supplementary material). On the contrary, the authors argue that this modelling of data augmentation captures typical real-world image transformations, such as "translations, crops, rotations, and [...] other pixel-wise augmentation methods such as sharpening, blurring, and color distortions" (second paragraph of Section 2). This is in sharp contrast with the limitations discussed by Rajput et al. (2019) themselves, on whose work the current paper takes inspiration for the theoretical analysis. In their Conclusion and Open Problems, Rajput et al. (2019) wrote: "There are several interesting open problems that we plan to tackle in the future. First, it would be interesting to theoretically analyze practical state-of-the-art augmentation methods, such as random crops, flips, and rotations. Such perturbations often fall outside our framework".

In line with Rajput et al. (2019), I would strongly argue that additive perturbations, while interesting from a mathematical point of view, only very weakly capture the extent of image transformations used in practice. Geometrical transformations can hardly be approximated by additive perturbations. One example of a widely common transformation that strongly differs from additive perturbations is horizontal flipping. This introduces a strong inductive bias, based on the properties of human visual perception, which is why it is included in almost all data augmentation schemes. Therefore, any conclusions from such a mathematical approximation to real-world data augmentation needs to be cautious. For example, the paper concludes saying that it has been shown "that data augmentation improves training and generalization by enlarging the smaller singular values of the neural network Jacobian". As argued above, this seems a strong claim given the limitations. The strength of the claims is particularly surprising after the authors wrote about other existing works on the theoretical analysis of data augmentation that they "do not provide insights on the effect of data augmentation on training deep neural networks".

My other main concern has probably a stronger impact on my overall assessment of the paper. This has to do with the motivation for the need for methods to select subsets of data on which to apply data augmentation. On the second sentence of the abstract, the authors write that "modern data augmentation techniques become computationally prohibitive for large datasets". This statement is extended in the introduction, where the authors cite several papers on so-called _automatic data augmentation_. First, these methods are indeed disproportionally expensive in computational terms, while providing only marginal improvements (in the best case) with respect to traditional, simple, cheap data augmentation techniques (Pérez and Wang, 2017). The authors mention that "multiple augmented examples are usually generated for a single data point to obtain better results, increasing the size of the training data by orders of magnitude", as though this was a weakness of data augmentation, while it is actually one of its main strengths. In particular, that the effective training size may be increased orders of magnitude while keeping the training time within the same order of magnitude.

An analysis of such advantages and the reasons for it (such as the fact that data augmentation can be performed in parallel to the parameter updates of the model, and even create a queue of data that would effectively keep the training time identical, given sufficient memory), as well as compelling evidence of the efficiency of data augmentation is provided by Hernández-García and König (2018). In that paper we see that training with light data augmentation (translations and horizontal flips) on the full training set provides large performance gains with a marginal increase of the training time. Furthermore, they also provide empirical evidence that training with 50 % data _and_ data augmentation (again on the full available set) achieves more than 95 % of the _full_ accuracy in about half the training time. Therefore, given that data augmentation can be applied almost _for free_ and training time can be traded by reducing the training data for a marginal reduction of the accuracy, why do we need a complex algorithm such as the one presented in this paper?

It is not clear from the paper how the training times are calculated, but we still see that in the best case the gains stay within the same order of magnitude. I believe that a stronger justification of the need for this method should take into account the considerations mentioned above, as well as a comparison with other alternative ways to trade training time for performance, such as changes in the architecture, etc.

### References

* Pérez and Wang. [The effectiveness of data augmentation in image classification using deep learning](https://arxiv.org/abs/1712.04621). 2017.
* Hernández-García and König. [Data augmentation instead of explicit regularization](https://arxiv.org/abs/1806.03852). 2018

**Summary Of The Paper:**

This paper first provides a theoretical analysis, under some assumptions, of the effect of data augmentation on the singular values of the network Jacobian and then proposes a method to improve the sample complexity of data augmentation, that is to select a subset of the data on which to perform the transformations to speed up training. The proposed method aims to find subsets of data whose augmentation yields similar alignment of the Jacobian with the residual vector as the fully augmented data.

**Summary Of The Review:**

While the paper seems solid in terms of correctness, I have a less positive impression due to, in my opinion, limited significance. This has to do with the simplification of data augmentation with respect to real world necessary for the theoretical analysis, as well as with the lack of strong motivation for the proposed algorithm.

---

> ### Author Response · Authors · 2021-11-12
> **Response to Reviewer fQLH**
>
> We thank the reviewer for their feedback. The reviewer raised two main concerns which we will address below. We believe the second concern is due to a misunderstanding of the strong augmentation methods we used in the paper. We clarify below:
>
> Limitations of additive perturbation model - Additive perturbations can in fact efficiently capture many (but not all) real-world image transformations, which Rajput et al. (2019) used in their analysis. However, as the reviewer and Rajput et al. pointed out, there are also inherent limitations to the additive perturbation model such as crops and flips. As such, we extended our theoretical analysis in Appendix B.3. by using general linear transformations that more effectively model such augmentations.
>
> Crucially, we would like to highlight that while our theoretical analysis in the main paper is based on the additive perturbation model, all the experimental results in Fig 2, 3 and the experiments section are based on a combination of crop, flip, rotation, translation, contrast, brightness, etc. Indeed, our empirical evaluations show that our theoretical results also hold for other types of augmentations that cannot be captured well by additive perturbations. We will further highlight this in the manuscript, and will leave the corresponding analysis to future work.
>
>
> We believe that there is a misunderstanding of the strong augmentation framework we are analyzing. The reviewer stated “the effective training size may be increased orders of magnitude while keeping the training time within the same order of magnitude.”, and that the “increasing size of training data by orders of magnitude” as a strength instead of a weakness. Given N training points, we believe that the reviewer understands the augmentation framework we use as creating (and training on) a new augmented dataset of size N. However, this is not the case. Strong augmentation methods such as [1] are crucial for getting the SOTA performance over existing light augmentations (which are already included by default in the baselines of [1] and our results). Such methods often add multiple data points to the training set. Taking the augmentation framework of [1] for example, given N training points per epoch, they generate C*N training examples where C=4 in their experimental setup, and then append them to the original dataset (and over here, all of these (C+1)*N training points are already transformed by light data augmentation by default). As such, training on these examples will require a much longer time (in particular, the running time increases C times without considering the time to generate the transformations). For such methods, generating the transformations in parallel to parameter update does not save much in terms of running time, as the main cost is training on the much larger (e.g. by 4x in [1]) augmented dataset. As such, we disagree with the statement from the reviewer that “data augmentation can be applied almost for free and training time can be traded by reducing the training data for a marginal reduction of the accuracy”. We believe that this statement is arised from a misunderstanding of our work.
>
> The reviewer also commented that “in the best case the gains stay within the same order of magnitude”, however this is also not true. For example, figure 4(a) and (b) shows that augmentation can provide 6.3x speedup on CIFAR10 while achieving 51% of the improvement from full data augmentation. To calculate the running time, as we discussed in Sec 5 under “Data and Augmentation”, we measured the average time taken for subset selection, gradient descent, and subset augmentation to compute total training time and speedups. Specifically, in Sec 5.1, the speedup is calculated as the ratio of the total running time for training on the full data and augmented subsets over the total running time for training on the full data and its augmentations. We will further clarify this in the manuscript.
>
> [1] Sen Wu, Hongyang Zhang, Gregory Valiant, and Christopher Ré. On the generalization effects of linear transformations in data augmentation. In International Conference on Machine Learning, pp. 10410–10420. PMLR, 2020.

---

> > ### Comment · Reviewer_fQLH · 2021-11-19
> > **Response to authors #1**
> >
> > Thank you for your responses. I appreciate the explanations but my main two concerns remain.
> >
> > Regarding the first point, the limitations of the additive perturbations model, I still believe this is in an important limitation of the theoretical analysis that is not discussed in sufficient depth in the paper and that limits the significance of the results. Furthermore, I disagree that the "empirical evaluations show that [the] theoretical results also hold for other types of augmentations that cannot be captured well by additive perturbations". The empirical results show demonstrate the application of the proposed algorithm, but does not the demonstrate the validity of the theoretical results on other kinds of transformations.
> >
> > This connects with my second point, which is to what extent we need such an algorithm. I humbly think there has not been a misunderstanding on my side in this regard. The argument you bring is that "strong augmentation methods such as [1] are crucial for getting the SOTA performance over existing light augmentations", and that such methods ([1]) require more computation because they generate C * N data points per epoch. I disagree that such methods are "crucial", as it has been demonstrated that large performance benefits are obtained by light augmentations that are computationally inexpensive. More sophisticated methods, or rather computationally more expensive methods, only provide marginal improvements in the best case, and it is an open question whether these improvements actually reflect better generalisation since we are already saturating the test sets where they are evaluated.
> >
> > To summarise, the gain in speed should be compared to standard (cheap) augmentation methods, not expensive methods like [1]. In fact, by looking at the results in the Appendix C, we see that such performance on CIFAR-10 (for instance) can be achieved with standard data augmentation methods.

---

### Official Review · Reviewer_xyXQ · 2021-11-03

**Correctness:** 3
**Technical Novelty And Significance:** 3
**Empirical Novelty And Significance:** 2
**Recommendation:** 5
**Confidence:** 3

**Main Review:**

Strengths:
* The paper presents a convincing reason why the approach can work in theory.
* The approach seems novel and connects various prior works.

Weaknesses:
* I thought the algorithm description in Section 4 was a bit rushed. For example, equation 11 was hard to understand and is core to the algorithm.
* I also though the Section 5 was rushed. The whole evaluation is ~1 page, which leaves the reader wanting for 1) how speedup is calculated 2) what is the objective of the experiments 3) what takeaways should be had. I would recommend moving other text to the appendix (e.g., anything reviewing prior work can be shortened and pushed to appendix).
* Figure 3b seems like a random walk. Figure 3c x-axis and text labels seem misaligned. Overall, I think this figure can be improved to give a clearer and more convincing story.
* The theory is appreciated but maybe it's worth contextualizing what values of the constants L and epsilon_0 are common in practice. Even a simple augmentation like translate can cause a large pixel deviation. For example, with 100 pixels horizontally, each varying by 1 intensity via a gradient (e.g., 0 to 99), shifting by 1 pixel will cause a change in the image that is 100 large.
* The good results seem very CIFAR10 specific. For example, Figure 4b shows SVNH has minimal improvement. More experimental evaluation is always good. For example, MNIST would have been an easy result to add and is in-fact mentioned in the paper, yet I didn't see it in evaluation.

**Summary Of The Paper:**

Deep learning uses augmentations to improve generalization performance. Using all augmentations for a dataset may slow down training. A subset selection technique is proposed (e.g., "coreset") using insights from the Neural Tangent Kernel (NMT) framework such that an alignment between the NMT Jacobian and the residuals is preserved. This alignment score is used to select the coreset using submodular optimization, which allows the model to be trained on a subset of augmented data (e.g., 0.1% to 30%) while preserving most augmentation benefits. The speedup of the method is reported to be up to 6.3x.

**Summary Of The Review:**

The paper is ok. I think there is enough theoretical justification for why the method could work, though I think empirical evidence is necessary to still show that. The theory is a bit disconnected and seems more like a synthesis of many different works; I wonder if it can be presented more succinctly? In any case, I found the writing good enough to follow. My biggest concern is the generalization of the evaluation, since it seems quite short at the moment and very CIFAR10 specific.

---

> ### Author Response · Authors · 2021-11-12
> **Response to Reviewer xyXQ**
>
> We thank the reviewer for acknowledging our theoretical contribution and the novelty of our work. We would like to highlight that further discussion of the experiments are omitted from the main text due to page limitation. However, we will address/clarify the following points in the revised version.
>
> Calculation of speedups: As we discussed in Sec 5 under “Data and Augmentation”, we measured the (average) time taken for subset selection, gradient descent, and subset augmentation to compute total training time and speedups. For example in Sec 5.1, the speedup is calculated as the ratio of the total running time for training on the full data and augmented subsets over the total running time for training on the full data and its augmentations.
>
> Experiment objectives: The experiments were aimed at showing that augmenting coresets are able to achieve larger speedup and stronger accuracy gains, compared to SoTA methods such as max-loss augmentation. To do so, we considered two scenarios: (1) In Sec. 5.1, we considered training on the full data while only augmenting the coreset elements. We showed in Table 1 and Fig. 4 that our method outperforms state-of-the-art max-loss augmentation strategy by 7.7% while achieving up to 6x speedup compared to training on the full data and its augmentations. We also showed that our method outperforms full data augmentation in presence of noisy labels. (2) In Sec 5.2, we considered training only on the coresets and their augmentations, to confirm that augmenting the coreset elements is more effective than augmenting random or max loss elements. Table 3 corroborates our theoretical results, by showing that augmenting coreset elements provides a much larger accuracy improvement compared to augmenting random or max loss examples. We will clarify this in the manuscript.
>
> Regarding MNIST and ImageNet: Improvement from full data augmentation is extremely small due to the already very high accuracy on MNIST. As such, most SoTA strong augmentation methods are not intended to be used for/ do not report results on MNIST [1,2,3]. Note that for the results reported in [4], the test sets are artificially augmented in certain ways in order to produce significant empirical evidence for improvement. We believe that this is not the right way to evaluate the performance. As such, we chose to not augment the test set in our experiment section. To strengthen our evaluation, we are currently running experiments on Places Dataset and ImageNet with ResNet56, which we will include in the revision if they complete in time. We note that the experiments conducted on WideResNet-28-10 in Table 3 using only 1% subset sizes on CIFAR10 already takes over a day to train on an A40 GPU. Running each experiment 5 times to obtain reliable results makes them even more computationally expensive, as such we initially chose to omit experiments on larger datasets like ImageNet, and instead increased complexity through noise and imbalance instead.
>
> Regarding augmentation model: For CIFAR10, we calculated $\epsilon_0$ (averaged over all the images) for the following transformations: Color transformation: 7.17, Brightness: 46.8, Sharpness: 5.1, Contrast: 22.7, horizontal-translation (max. 5%): 24.0, horizontal-translation (max. 10%): 38.1. For Color transformation, Brightness, Sharpness, and Contrast the corresponding parameter X is derived from a uniform distribution on the range [0.1, 1.9]. For example, for Color transformation X=0 represents a completely black image and X=1 represents the original image. Horizontal-translations are uniformly sampled from the range [-X%, X%]. We do agree that a perturbation model is not suitable for modelling certain kinds of augmentations - large translations, large rotations etc. in which $\epsilon_0$ will be high. For these cases, we instead analyze such augmentations more effectively as general linear transformations which we discuss in Appendix B.3.
>
> We thank the reviewer for their feedback!
>
> [1] Sen Wu, Hongyang Zhang, Gregory Valiant, and Christopher Ré. On the generalization effects of linear transformations in data augmentation. In International Conference on Machine Learning, pp. 10410–10420. PMLR, 2020.
>
> [2] Ekin D Cubuk, Barret Zoph, Dandelion Mane, Vijay Vasudevan, and Quoc V Le. Autoaugment: Learning augmentation strategies from data. In Proceedings of the IEEE/CVF Conference on Computer Vision and Pattern Recognition, pp. 113–123, 2019.
>
> [3] Ekin D Cubuk, Barret Zoph, Jonathon Shlens, and Quoc V Le. Randaugment: Practical automated data augmentation with a reduced search space. In Proceedings of the IEEE/CVF Conference on Computer Vision and Pattern Recognition Workshops, pp. 702–703, 2020.
>
> [4] Michael Kuchnik and Virginia Smith. Efficient augmentation via data subsampling. In International Conference on Learning Representations, 2018

---

> > ### Comment · Reviewer_xyXQ · 2021-11-29
> > **Response to authors**
> >
> > Thank you for the response!
> >
> > I am still borderline for my review. I appreciate the theory, but given that the analysis won't tightly predict behavior in the general case observed by the work, thorough empirical analysis is necessary (I would have liked Section 5 to be longer). I think there is sufficient depth in the individual experiments but the breadth of experiments is lacking. There are two ways to strengthen the experiments: 1) add datasets 2) add additional augmentations.
> >
> > The datasets are focused on CIFAR10. It's understandable that ImageNet is expensive to train, but it would be more convincing. An experiment on a smaller ImageNet variant would be a middle-ground. There are also other datasets that are smaller in size.
> >
> > For augmentations, the policy used: "We use the strong augmentation proposed by (Wu et al., 2020) to generate 4 distinct augmented examples by randomly sampling 2 augmentations from the same set used by (Cubuk et al., 2019; 2020) to apply to each example. A new set of default augmentations (random crop and horizontal flip) are also applied every R epochs to the original data" could be expanded. For example, the default augmentations seem peculiar to the setup. I think given that there is some disconnect between theory and practice, it would be interesting to analyze how the theory holds under controlled settings. As of now, there is a bit of a leap between theoretical understanding and the proposed experiments. This is a good place to analyze any differences between linear and non-linear models, as well, and further probing into the assumptions of NTK (e.g., wide nets).
> >
> > Unrelated to my review, but the organization of B.3 could be improved.

---

### Author Response · Authors · 2021-11-23
**Revision Uploaded**

Dear Reviewers,

We uploaded our rebuttal revision with minor modifications - correction of typos and updating Figure 4 as per recommendations of Reviewer 1cJe.

We note that applying strong augmentation to ImageNet is computationally very expensive - even without augmentation, training ImageNet on an NVIDIA M40 GPU with Resnet50 takes 2 weeks [1]. Adding in strong augmentation techniques and repeating experiments for 5 runs for each augmentation subset size and subset selection algorithm would take over 22 months to complete for a single NVIDIA M40 GPU. Instead, we are currently running experiments for both training and augmenting smaller subsets for larger-scale datasets such as ImageNet, and will report the results here and add them to the final version of our manuscript if they finish on time.

Thank you!

[1] Yang You, Zhao Zhang, C Hsieh, James Demmel, and Kurt Keutzer. Imagenet training in minutes. CoRR, abs/1709.05011, 2017b.

---

### Decision · Program_Chairs · 2022-01-20

**Decision:**

Reject

**Comment:**

The work presents a theoretical analysis of data augmentation, presenting evidence that data augmentation enlarges the smaller the singular values of the network Jacobian. Based on this theory the authors present a method for selecting a subset of training data to use with augmentation that decently approximates performance of training w/ augmentation on the full dataset. Reviewers overall agreed that the theoretical analysis was interesting, and did not find any flaws (though it is worth noting that the theory is restricted to additive perturbations). However, multiple reviewers found the presented experiments unconvincing, and questioned the stated motivation. The AC agrees with reviewers that most simple augmentations are not prohibitive in training speed. Certainly training on less data with a fixed epoch budget would require less compute time, but this is has nothing to do with augmentation and instead is a result of fewer steps taken in training. In the rebuttal, the authors argued that training on Imagenet is prohibitive with a single GPU (taking 2 weeks to do full training). However, given the authors claim their method speeds up training by a factor of 6.3x, then reducing ImageNet training from 2 weeks to 2 days would be a more convincing application of their method and would strengthen the work.